# On Universality of Deep Equivariant Networks

**Marco Pacini**[*]   **Mircea Petrache**[†]   **Bruno Lepri**[‡]   **Shubhendu Trivedi**[§]   **Robin Walters** [¶]

## Abstract

Universality results for equivariant neural networks remain rare. Those that do exist typically hold only in restrictive settings: either they rely on regular or higher-order tensor representations, leading to impractically high-dimensional hidden spaces, or they target specialized architectures, often confined to the invariant setting. This work develops a more general account. For invariant networks, we establish a universality theorem under separation constraints, showing that the addition of a fully connected readout layer secures approximation within the class of separation-constrained continuous functions. For equivariant networks, where results are even scarcer, we demonstrate that standard separability notions are inadequate and introduce the sharper criterion of *entry-wise separability*. We show that with sufficient depth or with the addition of appropriate readout layers, equivariant networks attain universality within the entry-wise separable regime. Together with prior results showing the failure of universality for shallow models, our findings identify depth and readout layers as a decisive mechanism for universality, additionally offering a unified perspective that subsumes and extends earlier specialized results.

## 1 Introduction

Symmetry has emerged as a key organizing principle in deep learning. Equivariant neural networks encode symmetry by ensuring that transformations of the input are mirrored by corresponding transformations of the output. This inductive bias has proven successful in applications ranging from vision and molecular modeling to representation learning on graphs and manifolds (Cohen & Welling, 2016; Kondor & Trivedi, 2018; Bronstein et al., 2021).

An important concern, however, is whether the introduced inductive biases may impose additional undesired constraints beyond symmetry. In this direction, the majority of work focuses on the study of *expressivity*, a broad notion that intuitively reflects the capacity of a family of models to represent or approximate arbitrarily complex target functions. This notion admits different formalizations, but two main approaches are currently investigated in the literature. The first directly tackles *universality*, understood here as the problem of approximating all symmetry-compatible target functions (Ravanbakhsh, 2020; Keriven & Peyré, 2019; Maron et al., 2019b; Sonoda et al., 2022), which, however, often requires models with impractically large intermediate representations. The second approach concerns the ability of models to distinguish input pairs, their *separation power*, an ability that also constrains the functions they can approximate. Separation has been extensively studied in the graph learning community through the lens of the Weisfeiler–Leman test (Morris et al., 2019; Maron et al., 2019a), and more recently in a general equivariant setting (Joshi et al., 2023; Pacini et al., 2025a). In particular, Chen et al. (2019) and Joshi et al. (2023) present initial approaches to universality that explicitly account for separation constraints. They establish universality results up to Weisfeiler–Leman and orbit separation, respectively, thereby furnishing the first cases of *separation-constrained universality*.

However, Pacini et al. (2025b) suggest a more nuanced landscape for the interaction between separation and universality. For instance, they present examples of invariant shallow architectures with the same separation power but different approximation power, showing that although separation

---
[*]University of Trento; Fondazione Bruno Kessler. `mpacini@fbk.eu`.
[†]PUC Chile. `mpetrache@uc.cl`.
[‡]Fondazione Bruno Kessler. `lepri@fbk.eu`
[§]`shubhendu@csail.mit.edu`
[¶]Northeastern University. `r.walters@northeastern.edu`

is a *necessary* condition for approximation, it may fail to be a *sufficient* one. Nevertheless, Zaheer et al. (2017), Qi et al. (2017), and Segol & Lipman (2020) show that adding fully connected readout layers or increasing the depth of this limited class of architectures transforms them into universal models up to separation. This suggests that depth and readout layers may play a crucial role in achieving separation-constrained universality and, more generally, in efficiently enhancing approximation power. In this paper, we shed light on this phenomenon by investigating the role of depth in separation-constrained universality, both in the invariant and equivariant regimes, and offer a unified framework that goes beyond earlier architecture-specific results. Our first result is a *separation-constrained universality theorem* for invariant networks, showing that models with fully connected readouts can approximate every continuous function consistent with their separation relation (Section 4). We then turn to the equivariant setting, where a simple example shows that standard separability is too coarse to characterize universality. To address this, we introduce the notion of *entry-wise separability* (Section 5.1). Intuitively, instead of considering the separation relation of the entire function, we examine all the separation relations of its projections onto individual output coordinates simultaneously. With this notion in place, we prove two *entry-wise separation-constrained universality theorems*. These results establish that deep equivariant networks achieve universality either when depth is sufficient to stabilize separation or when specific output layers act, in the equivariant setting, as surrogates of fully connected readouts in the invariant case (Section 5.2). In summary, our results identify depth and readouts as key factors for universality across broad classes of invariant and equivariant architectures. They clarify the role of separation in approximation and subsume earlier results restricted to shallow or architecture-specific settings.

We summarize our main contributions below:

- We establish a *separation-constrained universality theorem* for invariant networks (Theorem 1), showing that the addition of a fully connected readout guarantees approximation within the separation-constrained class.

- We introduce the concept of *entry-wise separability* and demonstrate, via Example 3, that standard separability fails to capture the universality class of equivariant networks.

- Building on this refinement, we prove two *entry-wise separation-constrained universality theorems*, showing that equivariant networks achieve universality either once entry-wise separation relations stabilize with depth (Theorem 2), or when equipped with specific readout layers (Theorem 3).

## 2 RELATED WORK

Equivariant architectures have emerged as a principled framework to incorporate symmetry into machine learning (Cohen & Welling, 2016; Kondor & Trivedi, 2018; Bronstein et al., 2021). Beyond convolution, a variety of techniques have been developed to enforce equivariance in over-parameterized or hierarchical representation-learning mechanisms, including approximate equivariance (Finzi et al., 2021; Petrache & Trivedi, 2023), tensor- and polynomial methods (Thomas et al., 2018), and hybrid polynomial models (Dym & Gortler, 2022). These approaches have been extended to different data structures, such as point clouds (Fuchs et al., 2020), graphs (Victor Garcia Satorras et al., 2021), and simplicial complexes (Battiloro et al., 2025). Thanks to this versatility, these models were able to adapt to diverse symmetry-sensitive domains, including high-energy physics (Bogatskiy et al., 2020), structural biology and drug discovery (Jumper et al., 2021), robotics (Huang et al., 2023), and medical imaging (Lafarge et al., 2021).

However, due to this heterogeneity of the landscape, a principled understanding of how such biases affect models remains fragmented and far from complete. Most work focuses on *expressivity*, the capacity of a model class to represent arbitrarily complex target functions. Ravanbakhsh (2020) and Sonoda et al. (2022) show that shallow architectures with regular hidden representations can approximate any equivariant map. However, these results require hidden spaces whose dimension scales with group size, making them impractical. Similarly, Maron et al. (2019b) establish universality for invariant networks with high-order tensor representations, though such constructions remain far from practical use. Among architectures used in practice, graph neural networks occupy a prominent role in the geometric deep learning literature. The study of their expressivity has been carried out primarily through the lenses of separation, via the Weisfeiler-Leman test, which enables fine-grained understanding of the model's separation capabilities. This focus is justified by the result of Chen et al.

(2019), who establish universality up to Weisfeiler–Leman separation. Building on this, Joshi et al. (2023) extended the result to geometric graph neural networks, establishing universality up to orbit separation. While these universality results remained confined to the invariant setting, the separation power of equivariant networks is now well characterized (Geerts, 2020; Geerts & Reutter, 2022; Pacini et al., 2025a), and depth plays a central role. However, Pacini et al. (2025b) recently showed that depth and additional readout layers play a more subtle role in approximation relative to separation, demonstrating that they can change the class of functions that are approximable without altering separation. This stands in stark contrast to the theory of classical neural networks, where depth is known to improve parameter efficiency but not the class of approximable functions (Telgarsky, 2016; Yarotsky, 2017; 2018).

We extend this literature in two ways. First, we prove that adding a fully connected readout layer is sufficient to achieve separation-constrained universality for invariant neural networks. Second, we introduce and analyze *entry-wise separability*, showing that it provides the appropriate refinement for extending separation-constrained universality to equivariant networks. We present these results in a mathematical framework that is general enough to encompass prior settings while precisely capturing the underlying phenomena.

## 3 Preliminaries

### 3.1 Groups and Equivariance

We study functions that behave consistently under prescribed transformations. Among them, some are naturally formalized by the algebraic structure of a *group*: sets of transformations closed under composition and containing inverses and an identity. While group theory provides the natural framework for reasoning about symmetry, in the context of neural networks it is convenient to reformulate these ideas in linear-algebraic terms. This is achieved through *representation theory*, which encodes abstract group elements as matrix actions on vector spaces (Serre, 1977).

*Permutation representations* play a central role in this work. They arise when a group $G$ acts on a finite set $X$, where the action is given by an identification of $G$ as a subset of permutations of $X$. Let $\mathbb{R}^X$ denote the space of real-valued functions on $X$, and for each $x \in X$ define the indicator $e_x \in \mathbb{R}^X$ by $e_x(x) = 1$ and $e_x(y) = 0$ for $y \neq x$. The collection $\{e_x\}_{x \in X}$ forms a canonical basis of $\mathbb{R}^X$. The associated permutation representation is the linear action on $V = \mathbb{R}^X$ given by $g(e_x) = e_{gx}$, where $gx$ is the result of $g$ acting on $x$ for $g \in G, x \in X$.

If $V$ and $W$ are permutation representations of $G$, a map $\phi : V \to W$ is called *G-equivariant* when $\phi(gv) = g\phi(v)$ for all $g \in G$, $v \in V$. We denote by $\mathrm{Hom}(V, W)$ the space of linear maps and by $\mathrm{Hom}_G(V, W)$ the subspace of $G$-equivariant linear maps. Similarly, $\mathrm{Aff}(V, W)$ denotes the space of affine maps and $\mathrm{Aff}_G(V, W) \subseteq \mathrm{Aff}(V, W)$ the subset of $G$-equivariant affine maps. All these spaces are real vector spaces under pointwise addition and scalar multiplication.

Further preliminaries are provided in Appendix A.

### 3.2 Layer Spaces, Neural Spaces & Equivariant Neural Networks

We now describe equivariant neural networks, which are model classes with group equivariant layers. Throughout, we restrict attention to networks equivariant under the action of a finite group, with layers given by permutation representations and equipped with arbitrary point-wise continuous activations. We begin by introducing the notion of a *layer space*, namely a space of affine maps subject to additional constraints—such as equivariance requirements or restrictions on the set of permissible filters—which will serve as the fundamental building block of the neural architectures under consideration.

**Definition 1** (Layer Spaces). *Let $G$ be a finite group acting on a finite set $X$, let $\mathbb{R}^X$ be the permutation representation associated with this action, and let $V$ be another permutation representation of $G$. A* layer space *is a subset $M \subseteq \mathrm{Aff}_G(V, \mathbb{R}^X)$. In this work, we focus on spaces of the form*

$$M = \left\{ v \mapsto \sum_{i=1}^{k} x_i \, \phi^i(v) + \sum_{j=1}^{\ell} y_j \, \mathbb{1}_{X_j} \ \Big| \ x_1, \ldots, x_k, y_1, \ldots, y_\ell \in \mathbb{R} \right\}, \tag{1}$$

where $\phi^1, \ldots, \phi^k \in \mathrm{Hom}_G(V, \mathbb{R}^X)$, the sets $X_1, \ldots, X_\ell$ denote all the orbits of $X$ under the $G$-action, and $\mathbb{1}_{X_j} := \sum_{x \in X_j} e_x$ for $j = 1, \ldots, \ell$, with $\{e_x\}_{x \in X}$ the canonical basis of $\mathbb{R}^X$.

**Example 1.** *We give some examples of layer spaces, $L$, $I$, $C$, and $P$, which will be used throughout the manuscript as running references in our analysis of the universality phenomena. These layer spaces correspond to widely used architectures in geometric deep learning and illustrate how standard models naturally fit into the general form (1).*

   *(i)* ***Linear Layer:*** *Linear layers in standard neural networks are given by elements of $\mathrm{Aff}(\mathbb{R}^n, \mathbb{R}^m)$. For the action of any group, we can define the set of affine linear maps between trivial representations $L := \mathrm{Aff}(\mathbb{R}, \mathbb{R})$, whose relevance will become clearer later, for instance, in relation to (3).*

   *(ii)* ***Invariant Layer:*** *Let $G$ be a finite group acting on a finite set $X$, and let $\mathbb{R}^X$ denote the associated permutation representation. We denote by $\mathbb{R}$ the trivial real representation of $G$. The space of $G$-invariant affine maps from $\mathbb{R}^X$ to $\mathbb{R}$ is denoted by $I := \mathrm{Aff}_G(\mathbb{R}^X, \mathbb{R})$. If $X = X_1 \sqcup \cdots \sqcup X_\ell$ is the orbit decomposition of $X$, then we have the characterization*

$$I := \Big\{ v \mapsto \sum_{i=1}^{\ell} x_i \mathbb{1}_{X_i}^\top \cdot v + y \mid x_1, \ldots, x_\ell, y \in \mathbb{R} \Big\}.$$

   *(iii)* ***Convolutional Layer:*** *Standard convolutional layers correspond to maps equivariant with respect to the cyclic group $G = \mathbb{Z}_n \times \mathbb{Z}_n$, acting by the standard cyclic permutations on the product $X = [n] \times [n]$, and can be naturally formulated within the framework of permutation representations. Here we consider the generalization to general finite $G$ acting on finite $X$, and focus on convolutional layers with filter width $1$ between general permutation representations $\mathbb{R}^X$. These can be written in the form of (1) as follows.*

$$C := \Big\{ v \mapsto x\,\mathrm{id} \cdot v + \sum_{i=1}^{\ell} y_i \mathbb{1}_{X_i} \,\Big|\, x, y_1, \ldots, y_\ell \in \mathbb{R} \Big\}. \tag{2}$$

   *Note that here $C \subseteq \mathrm{Aff}_G(\mathbb{R}^X, \mathbb{R}^X)$ for any action of $G$ on $X$, where $X = X_1 \sqcup \cdots \sqcup X_\ell$ denotes the orbit decomposition of $X$. The same setting can be extended to wider filters, but for ease of exposition, will not be used in this paper.*

   *(iv)* ***PointNet Layer:*** *Sum-pooling PointNet layers (Qi et al., 2017) are designed to process unordered collections, such as point clouds, by enforcing permutation equivariance. In the simplest case, an input configuration of $n$ real elements is represented as a vector $a \in \mathbb{R}^n$, where we identify $\mathbb{R}^X = \mathbb{R}^{[n]} \cong \mathbb{R}^n$. This definition extends analogously to the general case with multi-dimensional features. Equivariant PointNet layers act on such inputs using maps in the space $\mathrm{Aff}_{S_n}(\mathbb{R}^n, \mathbb{R}^n)$. Zaheer et al. (2017) characterized this space as*

$$P := \Big\{ v \mapsto (x_1\,\mathrm{id} + x_2 \mathbb{1}\mathbb{1}^\top) \cdot v + y\mathbb{1} \,\Big|\, x_1, x_2, y \in \mathbb{R} \Big\},$$

   *where $\mathbb{1} = \mathbb{1}_{[n]} = [1, \ldots, 1]^\top$.*

We restrict our study to point-wise activations, also referred to in the literature as *component-wise* or *entry-wise* activations.

**Definition 2** (Point-wise Activation)**.** *Let $\sigma : \mathbb{R} \to \mathbb{R}$ be a nonlinear activation. Given a permutation representation $\mathbb{R}^X$ of a group $G$, we define the associated point-wise activation $\tilde{\sigma} : \mathbb{R}^X \to \mathbb{R}^X$ by $\tilde{\sigma}\big(\sum_{x \in X} \alpha_x e_x\big) = \sum_{x \in X} \sigma(\alpha_x) e_x$. Wherever the usage is unambiguous from context, we will denote both $\sigma$ and $\tilde{\sigma}$ by the same symbol.*

Throughout the paper, we assume that the activation function $\sigma : \mathbb{R} \to \mathbb{R}$ is non-polynomial, as we focus on universality results for architectures of fixed depth.

We now state the definition of a neural network and of the functional space of a fixed neural architecture, which we call a *neural space*, also referred to in the literature as a *neuromanifold* (Calin, 2020).

**Definition 3** (Neural Networks and Neural Spaces). *Let $G$ be a group and $V_0, \ldots, V_d$ be permutation representations of $G$. For each $i = 1, \ldots, d$, let $M_i$ be a layer space in $\mathrm{Aff}_G(V_{i-1}, V_i)$. For $d \geq 2$, the* neural space *associated with layers $M_1, \ldots, M_d$ and activation $\sigma$ is defined recursively by*

$$\mathcal{N}_\sigma(M_1, \ldots, M_d) = \left\{ \phi^d \circ \tilde{\sigma} \circ \cdots \circ \tilde{\sigma} \circ \phi^1 \mid \phi^i \in M_i \ \text{for each } i = 1, \ldots, d \right\}.$$

*Any $\eta^d \in \mathcal{N}_\sigma(M_1, \ldots, M_d)$ is called a* neural network *with layers in $M_1, \ldots, M_d$ and activation $\sigma$.*

### 3.3 Universality Classes and Separation

We aim to characterize the class of continuous functions approximable by neural networks with fixed architecture. We generalize the notion of *universality class* introduced by Pacini et al. (2025b) for shallow networks, to networks of arbitrary depth. Before giving a formal definition, we introduce an auxiliary notion which plays the role of width in classical (non-equivariant) universality results (Pinkus, 1999), since it can be interpreted as the dimension of intermediate invariant feature representations. If $V$ and $W$ are permutation representations of a finite group $G$ and $M \subseteq \mathrm{Aff}_G(V, W)$ is as defined in (1), then, for each $h, k \in \mathbb{N}$ we define $M^{k \times h}$ as the subspace of $\mathrm{Aff}_G(V \otimes \mathbb{R}^k, W \otimes \mathbb{R}^h)$:

$$M^{k \times h} := \left\{ \begin{array}{c} (x_1, \ldots, x_k) \mapsto \left( \sum_{j=1}^k f_{1,j}(x_j), \ldots, \sum_{j=1}^k f_{h,j}(x_j) \right) \\ f_{ij} \in M, i = 1, \ldots, h, j = 1, \ldots, k \end{array} \right\}. \tag{3}$$

**Example 2.** *Recall that the definition of $L = \mathrm{Aff}(\mathbb{R}, \mathbb{R})$, the layer space $L^{k \times h}$ is the set of all affine maps from $\mathbb{R}^k$ to $\mathbb{R}^h$, namely $\mathrm{Aff}(\mathbb{R}^k, \mathbb{R}^h)$. Since $L = \mathrm{Aff}_G(\mathbb{R}, \mathbb{R})$ where $G$ acts trivially on $\mathbb{R}$, this layer space can be interpreted as the space of affine $G$-equivariant maps between trivial representations. In this sense, the multiplicities $k$ and $h$ correspond to the widths of intermediate representations in the standard neural network setting.*

With the above notation in place, we can now provide a general definition of universality classes. Intuitively, a universality class consists of all functions that can be uniformly approximated on compact sets by neural networks of a given architecture, with variable multiplicities of layer spaces.

**Definition 4** (Universality Classes). *The* universality class *$\mathcal{U}_\sigma(M_1, \ldots, M_d)$ associated with the layer spaces $M_1, \ldots, M_d$ is defined as*

$$\mathcal{U}_\sigma(M_1, \ldots, M_d) := \overline{\bigcup_{\vec{h} \in \mathbb{N}^{d-1}} \mathcal{N}_\sigma\left( M_1^{1 \times h_1}, M_2^{h_1 \times h_2}, M_3^{h_2 \times h_3}, \ldots, M_{d-1}^{h_{d-2} \times h_{d-1}}, M_d^{h_{d-1} \times 1} \right)},$$

*where the overline denotes closure in the topology of uniform convergence on compact sets.*

Invariant networks are inherently unable to distinguish between elements in the same group orbits, but additional undesired separability constraints may arise when dealing with neural networks with particular architectures employed in practice. A prominent example is given by graph neural networks, which are known to be subject to separation constraints equivalent to the Weisfeiler–Leman test (Chen et al., 2019). To study the universality classes arising from architectures employed in practice, we must therefore take these separability constraints into account. We will use the following natural definitions of *separation* and of *separation-constrained universality*.

**Definition 5** (Separation-Constrained Universality). *Let $\mathcal{U} \subseteq \{ f : V \to W \}$ be a family of functions. We say that $\mathcal{U}$ **separates** two points $\alpha, \beta \in V$ if there exists $f \in \mathcal{U}$ such that $f(\alpha) \neq f(\beta)$. The set of pairs that cannot be distinguished by any $f \in \mathcal{U}$ induces an equivalence relation:*

$$\rho(\mathcal{U}) = \left\{ (\alpha, \beta) \in V \times V \mid f(\alpha) = f(\beta) \ \text{for all } f \in \mathcal{U} \right\}.$$

*We say that $\mathcal{U}$ is **separation-constrained universal** if it approximates exactly the class of continuous functions that preserve the equivalence relation $\rho = \rho(\mathcal{U})$, namely*

$$\mathcal{C}_\rho(V, W) = \left\{ f \in \mathcal{C}(V, W) \mid f(\alpha) = f(\beta) \ \text{whenever } (\alpha, \beta) \in \rho \right\}.$$

Note that separability is a *necessary condition* for uniform approximation on compact sets: any sequence of functions with prescribed separation power $\rho$ converges only to functions that also respect $\rho$. In other words, $\mathcal{C}_\rho(V, W)$ is a closed subset of $\mathcal{C}(V, W)$ in the topology of uniform convergence on compact sets.

As noted in Sections 1 and 2, the literature on universality for equivariant neural networks is typically architecture-dependent, often focusing on the invariant case, and when general, relying on impractically large intermediate representations.

**Prior Work.** *Here we summarize, to the best of our knowledge, known universality results, recasting them within a unified framework of universality classes and separation-constrained approximability.*

1. *The classical universality theorem of Pinkus (1999), which states that neural networks can approximate any continuous function, translates in this framework as $\mathcal{U}_\sigma(L, L) = \mathcal{C}(\mathbb{R}, \mathbb{R})$.*

2. *Segol & Lipman (2020) show that a simplified version of 3-layer PointNets, where convolutional filters of width 1 appear only in certain layers (see Examples 1.iii and iv), is universal. This, in turn, implies that full 3-layer PointNets are universal in the class of continuous $S_n$-equivariant functions. Namely, $\mathcal{U}_\sigma(C, P, C) = \mathcal{U}_\sigma(P, P, P) = \mathcal{C}_{S_n}(\mathbb{R}^n, \mathbb{R}^n)$.*

3. *Ravanbakhsh (2020) shows that shallow equivariant networks with regular representations as hidden layers are universal among $G$-equivariant functions. Namely, for permutation representations $V$ and $W$, $\mathcal{U}_\sigma(M, N) = \mathcal{C}_G(V, W)$ where $M = \mathrm{Aff}_G(V, \mathbb{R}^G)$ and $N = \mathrm{Aff}_G(\mathbb{R}^G, W)$.*

4. *Joshi et al. (2023) show that models expressive enough to distinguish all $G$-orbits become universal in the invariant sense once augmented with a shallow neural network head. Namely, if the neural space $\mathcal{N}_\sigma(M_1, \ldots, M_d, I)$ separates $G$-orbits in $\mathbb{R}^n$, then the associated universality class, augmented with a shallow network head, satisfies $\mathcal{U}_\sigma(M_1, \ldots, M_d, I, L) = \mathcal{C}_G(\mathbb{R}^n, \mathbb{R})$.*

5. *Geerts (2020); Maron et al. (2019a); Chen et al. (2019) show that graph neural networks and invariant graph networks (Maron et al., 2018) can approximate any continuous invariant function with the same separation power as the Weisfeiler–Leman test. Namely, for the layer space $M = \mathrm{Aff}_{S_n}((\mathbb{R}^n)^{\otimes k}, (\mathbb{R}^n)^{\otimes k})$, which processes $k$-order relational structures equivariantly, $\mathcal{U}_\sigma(\underbrace{M, \ldots, M}_{d \text{ times}}, I, L) = \mathcal{C}_{k\text{-WL}_d}((\mathbb{R}^n)^{\otimes k}, (\mathbb{R}^n)^{\otimes k})$, that is, the set of continuous functions with the same separation power as the $k$-WL test after $d$ iterations.*

6. *Moreover, Pacini et al. (2025b) show that in some cases the final trivial layer, as in the two previous examples, is necessary for separation-constrained universality when certain representations are involved. Namely, they prove that $\mathcal{U}_\sigma(C, I) \subsetneq \mathcal{U}_\sigma(P, I) \subsetneq \mathcal{C}_{S_n}(\mathbb{R}^n, \mathbb{R})$, even though these spaces exhibit the same separation power: $\rho(\mathcal{U}_\sigma(C, I)) = \rho(\mathcal{U}_\sigma(P, I)) = \rho(\mathcal{C}_{S_n}(\mathbb{R}^n, \mathbb{R}))$.*

## 4 SEPARATION-CONSTRAINED UNIVERSALITY FOR INVARIANT NETWORKS

In this section, we establish a general result on separation-constrained universality (Definition 5) for invariant neural networks, extending prior works on invariant universality (see Prior Work 4 and 5). In particular, we prove that pathological mismatches between separation power and approximation power (see Prior Work 6) can always be resolved by adding a fully connected readout layer.

**Theorem 1.** *Let $M_1, \ldots, M_d$ be layer spaces as defined in Definition 1 and recall that $I$ denotes a layer space of invariant linear functions, namely, a subset of the layer space from Example 1.ii. Set $\rho = \rho(\mathcal{U}_\sigma(M_1, \ldots, M_d, I))$. Then*

$$\mathcal{U}_\sigma(M_1, \ldots, M_d, I, L) = \mathcal{C}_\rho(V). \tag{4}$$

*Proof of Theorem 1.* Note that by Theorem 4 in Pacini et al. (2025a) and the remark following it, $\rho$ is preserved under the extension from $\mathcal{N}_\sigma(M_1, \ldots, M_d, I, L)$ to $\mathcal{U}_\sigma(M_1, \ldots, M_d, I, L)$ and from $\mathcal{N}_\sigma(M_1, \ldots, M_d, I)$ to $\mathcal{U}_\sigma(M_1, \ldots, M_d, I)$, therefore

$$\rho = \rho(\mathcal{N}_\sigma(M_1, \ldots, M_d, I)) = \rho(\mathcal{N}_\sigma(M_1, \ldots, M_d, I, L)).$$

Hence we get $\mathcal{U}_\sigma(M_1, \ldots, M_d, I, L) \subseteq \mathcal{C}_\rho(V)$ and we only have to prove the opposite inclusion. Given functions $f_1, \ldots, f_h \in \mathcal{C}(V, \mathbb{R})$, define their parallelization as $F_h = (f_1, \ldots, f_h) \colon V \longrightarrow \mathbb{R}^h$, $F_h(x) = (f_1(x), \ldots, f_h(x))$, and set

$$\mathcal{A}_h := \left\{ \eta \circ F_h \mid \eta \in \mathcal{C}(\mathbb{R}^h) \right\}, \quad \mathcal{A}'_h := \left\{ \eta \circ F_h \mid \eta \in \bigcup_{k \in \mathbb{N}} \mathcal{N}_\sigma(L^{h \times k}, L^{k \times 1}) \right\}. \tag{5}$$

Note that by the universal approximation theorem, $\mathcal{A}_h = \overline{\mathcal{A}_h} = \overline{\mathcal{A}'_h}$. From now on we will take $\mathcal{F} = \{f_h\}_{h \in \mathbb{N}}$ to be a family of functions such that $\rho(\mathcal{F}) = \rho$. We get via a result from the appendix that

$$\mathcal{C}_\rho(V) \overset{\text{Lemma 3}}{=} \overline{\bigcup_{h \in \mathbb{N}} \mathcal{A}_h} = \overline{\bigcup_{h \in \mathbb{N}} \mathcal{A}'_h}. \tag{6}$$

Define

$$\mathcal{N}_h := \bigcup_{\vec{k} \in \mathbb{N}^{d+1}} \mathcal{N}_\sigma(M_1^{1 \times k_1}, \ldots, M_d^{k_{d-1} \times k_d}, I^{k_d \times h}) \quad \text{for each } h \in \mathbb{N}.$$

Then we can write

$$\mathcal{U}_\sigma(M_1, \ldots, M_d, I, L) = \overline{\bigcup_{\vec{k} \in \mathbb{N}^{d+1}} \mathcal{N}_\sigma(M_1^{1 \times k_1}, \ldots, M_d^{k_{d-1} \times k_d}, I^{k_d \times k}, L^{k \times 1})}$$

$$= \overline{\bigcup_{\tilde{k} \in \mathbb{N}^{d+2}} \mathcal{N}_\sigma(M_1^{1 \times k_1}, \ldots, M_d^{k_{d-1} \times k_d}, I^{k_d \times h}) \,\hat{\circ}\, \mathcal{N}_\sigma(L^{h \times k}, L^{k \times 1})}$$

$$= \overline{\bigcup_{h \in \mathbb{N}} \left\{ \eta \circ f \mid f \in \mathcal{N}_h, \, \eta \in \bigcup_{k \in \mathbb{N}} \mathcal{N}_\sigma(L^{h \times k}, L^{k \times 1}) \right\}}$$

$$\overset{\text{Equation 5}}{\supseteq} \overline{\bigcup_{h \in \mathbb{N}} \mathcal{A}'_h} \overset{\text{Equation 6}}{=} \mathcal{C}_\rho(V).$$

To prove the above inclusion, if $f_1, \ldots, f_h \in \mathcal{N}_\sigma(M_1, \ldots, M_d, I)$ then their parallelization $(f_1, \ldots, f_h)$ belongs to $\mathcal{N}_h$ by Lemma 1 from the appendix. The last equality holds because of Equation 6 and Corollary 2, since there exists a family of networks $\mathcal{F} = \{f_h\}_{h \in \mathbb{N}}$ such that $f_h \in \mathcal{N}_\sigma(M_1, \ldots, M_d, I)$ for each $h \in \mathbb{N}$ and $\rho(\mathcal{F}) = \rho$, and we can use this family to define $\mathcal{A}'_h$. $\qquad\square$

## 5 UNIVERSALITY OF EQUIVARIANT NEURAL NETWORKS

In this section, we extend the previous results to the equivariant setting. However, important differences between the invariant and equivariant cases emerge: in Section 5.1 we show that the standard form of the separation relation as in Definition 5 often fails to faithfully characterize equivariant universality classes, requiring us to introduce the notion of entry-wise separation (Definition 6). In Section 5.2, we establish universality theorems analogous to Theorem 1, showing that the outcome crucially depends on the choice of output space.

### 5.1 ENTRY-WISE SEPARATION

Here, we study equivariant functions by reducing the problem to the analysis of suitable invariant functions, thereby connecting our setting to the results of Section 4. The main tool for this reduction is the projection onto output coordinates. More precisely, let $G$ be a finite group acting on the finite set $X$. For $x \in X$ consider the stabilizer of $x$, given by $G_x = \mathrm{Stab}_G(x) := \{g \in G \mid gx = x\}$, and the linear projection $\pi_x : \mathbb{R}^X \to \mathbb{R}$ onto the $x$-th coordinate. Then $\pi_x$ induces the pushforward map

$$\pi_{x*} : \mathcal{C}_G(V, \mathbb{R}^X) \longrightarrow \mathcal{C}_{G_x}(V)$$
$$f \longmapsto \pi_x \circ f.$$

Since the vector of projections satisfies $(\pi_x)_{x \in X} = \mathrm{id}_{\mathbb{R}^X}$, it follows that $(\pi_{x*})_{x \in X}$ acts as the identity on $\mathcal{C}_G(V, \mathbb{R}^X)$. Thus, the study of universality for equivariant maps reduces to the problem of synchronous universality of the invariant projection maps. However, below Proposition 1 shows that the interaction between equivariance and the global separation $\rho$ is non-trivial when projecting functions onto different output entries.

**Proposition 1.** *Let $\rho = \rho(\mathcal{N})$ be the separation relation of a family of equivariant neural networks $\mathcal{N}$. The restriction of $\pi_x$ to*

$$\mathcal{C}_{G,\rho}(V, \mathbb{R}^X) := \mathcal{C}_G(V, \mathbb{R}^X) \cap \mathcal{C}_\rho(V, \mathbb{R}^X)$$

*is surjective onto $\mathcal{C}_{G_x,\rho}(V)$, the space of $G_x$-invariant functions with separation relation $\rho$.*

The proof for Proposition 1 and of all subsequent results may be found in the Appendix.

Proposition 1 shows that, after projection onto a single output coordinate, the space of equivariant functions with separation $\rho$ is constrained by a stricter relation. This relation combines $\rho$ with the $G_x$-invariance relation, which identifies elements within each $G_x$-orbit. However, the following example shows that this stricter condition remains insufficient to correctly characterize the universality classes associated with equivariant architectures.

**Example 3** (Separation for CNNs). *Let $C$ be the layer space of convolutional filters with width* 1 *defined in Example 1.iii. For the purpose of this example, it is sufficient to restrict $C$ to go from $\mathbb{R}^n$ to $\mathbb{R}^n$ with $S_n$ acting in the standard way on $\mathbb{R}^n$. Hence, (2) becomes*

$$C := \left\{ v \mapsto x \operatorname{id} \cdot v + y \mathbb{1} \ \middle| \ x, y \in \mathbb{R} \right\}.$$

*Consider the universality class for $d \geq 2$:*

$$\mathcal{U}_\sigma^d(C) := \mathcal{U}_\sigma(\underbrace{C, \ldots, C}_{d \text{ times}}).$$

*We can show (see Proposition 5) that*

$$\mathcal{U}_\sigma^d(C) = \{(x_1, \ldots, x_n) \mapsto (f(x_1), \ldots, f(x_n)) \mid f \in \mathcal{C}(\mathbb{R})\} \subsetneq \mathcal{C}_{S_n}(\mathbb{R}^n, \mathbb{R}^n). \tag{7}$$

*Note that $\operatorname{id}_{\mathbb{R}^n} \in \mathcal{U}_\sigma^d(C)$. Then, $\rho(\mathcal{U}_\sigma^d(C))$ is the trivial separation relation, namely $\rho(\mathcal{U}_\sigma^d(C)) = \{(x, x) \mid x \in \mathbb{R}^n\}$. Thus, the target space of separation-constrained universality is $\mathcal{C}_{S_n, \rho}(\mathbb{R}^n, \mathbb{R}^n) = \mathcal{C}_{S_n}(\mathbb{R}^n, \mathbb{R}^n)$. However, (7) shows that $\mathcal{U}_\sigma^d(C) \subsetneq \mathcal{C}_{S_n}(\mathbb{R}^n, \mathbb{R}^n)$ for each $d \geq 2$. Or equivalently, in this case separation-constrained universality can never be attained, regardless of depth $d$.*

Example 3 shows that characterizing equivariant universality classes requires a finer notion of separability, which we now define.

**Definition 6** (Entry-wise Separation). *Let $G$ be a finite group acting on a finite set $X = \{x_1, \ldots, x_n\}$, and let $\mathbb{R}^X$ denote the associated permutation representation. Let $V$ be another permutation representation over $G$ and $\mathcal{N}$ a neural space of functions in $\mathcal{C}_G(V, \mathbb{R}^X)$. Let $\pi_x : \mathbb{R}^X \to \mathbb{R}$ be the linear projection onto the $x$-th component for each $x \in X$. Define the family of separation relations*

$$\rho_x(\mathcal{N}) := \{(\alpha, \beta) \in V \times V \mid \pi_x f(\alpha) = \pi_x f(\beta) \text{ for all } f \in \mathcal{N}\}.$$

*for each $x \in X$. We define the* entry-wise separation relation *as the collection of separation relations*

$$\overline{\rho}(\mathcal{N}) = \big(\rho_{x_1}(\mathcal{N}), \ldots, \rho_{x_n}(\mathcal{N})\big).$$

*We define the set of continuous functions that respect $\overline{\rho}$ as*

$$\mathcal{C}_{\overline{\rho}}(V, \mathbb{R}^X) := \big\{ f \in \mathcal{C}(V, \mathbb{R}^X) \mid \pi_x f(v_1) = \pi_x f(v_2) \ \forall (v_1, v_2) \in \rho_x(\mathcal{N}), \forall x \in X \big\}.$$

*If a universality class with entry-wise separation $\overline{\rho}$ coincides with $\mathcal{C}_{\overline{\rho}}$, we call it **entry-wise separation universal**.*

Note that $\rho(\mathcal{N}) = \rho_{x_1}(\mathcal{N}) \cap \cdots \cap \rho_{x_n}(\mathcal{N})$, so the standard separation relation is implied by the entry-wise separation relations. That is $\mathcal{N} \subseteq \mathcal{C}_{\overline{\rho}(\mathcal{N})}(V, \mathbb{R}^X) \subseteq \mathcal{C}_{\rho(\mathcal{N})}(V, \mathbb{R}^X)$. As noted in Section 3.3, separation is a necessary condition for approximation, and now we see entry-wise separation is necessary as well. Note that in certain cases entry-wise separation reduces entirely to the standard separation relation, for instance in the invariant case where $G$ acts trivially on $\mathbb{R}$, or more simply when $\rho(\mathcal{N}) = \rho_{x_1}(\mathcal{N}) = \cdots = \rho_{x_n}(\mathcal{N})$. Yet, Example 3 shows that entry-wise separation can, in fact, be strictly stronger than standard separation. Indeed, on the one hand (7) gives

$$\pi_{1*}\mathcal{U}_\sigma^d(C) = \{ (x_1, \ldots, x_n) \mapsto f(x_1) \mid f \in \mathcal{C}(\mathbb{R}) \} \subsetneq \mathcal{C}_{\operatorname{Stab}_{S_n}(1)}(\mathbb{R}^n, \mathbb{R}),$$

while on the other hand, we have $\pi_{1*}\mathcal{U}_\sigma^d(C) = \mathcal{C}_{\rho_1}(\mathbb{R}^n, \mathbb{R})$. If we denote $\mathbb{R}^n = \mathbb{R} \times \mathbb{R}^{n-1}$, here $\rho_1 := \big\{ ((x_1, \overline{x}), (y_1, \overline{y})) \in (\mathbb{R} \times \mathbb{R}^{n-1})^2 \ \big| \ x_1 = y_1 \big\}$. Analogous results hold for the other $\rho_i$, with $i = 2, \ldots, n$. This proves the following proposition and shows that the universality class in Example 3 can be completely characterized by entry-wise separation universality.

**Proposition 2.** *Define $\overline{\rho} = \overline{\rho}\big(\mathcal{U}_\sigma^d(C)\big)$. Then, $\mathcal{U}_\sigma^d(C) = \mathcal{C}_{\overline{\rho}}(\mathbb{R}^n, \mathbb{R}^n)$.*

## 5.2 Entry-wise Separation Constrained Universality

Now we are ready to state universality results under the more general notion of entry-wise separability as discussed in Section 5.1.

**Theorem 2.** *Let $V_0, \ldots, V_h$ be permutation representations of a finite group $G$. Let $X$ be a finite $G$-set and $\mathbb{R}^X$ its associated permutation representation. Let $M_1, \ldots, M_f$ be layer spaces in $\mathrm{Aff}_G(V_{i-1}, V_i)$ for $i = 1, \ldots, f$, and let $M$ be a layer space in $\mathrm{Aff}_G(\mathbb{R}^X, \mathbb{R}^X)$ containing the identity map. Let $d$ be such that*

$$\overline{\rho} := \overline{\rho}\big(\mathcal{U}_\sigma(M_1, \ldots, M_f, \underbrace{M, \ldots, M}_{d \text{ times}})\big) = \overline{\rho}\big(\mathcal{U}_\sigma(M_1, \ldots, M_f, \underbrace{M, \ldots, M}_{d+1 \text{ times}})\big). \tag{8}$$

*Then,*

$$\mathcal{U}_\sigma(M_1, \ldots, M_f, \underbrace{M, \ldots, M}_{d+1 \text{ times}}) = \mathcal{C}_{\overline{\rho}}(V_0, \mathbb{R}^X).$$

*In other words, repeating the output layer beyond the separation-stabilization threshold ensures entry-wise separation-constrained universality.*

Since by Theorem 3 of Pacini et al. (2025a), separation is known to stabilize after a certain depth, we obtain the following corollary.

**Corollary 1.** *Assume the notation of Theorem 2. There exists a natural number $D$ for which $\mathcal{U}_\sigma(M_1, \ldots, M_f, \underbrace{M, \ldots, M}_{d \text{ times}})$ is entry-wise separation-constrained universal for each $d \geq D$.*

In a different direction, we can show that entry-wise separation-constrained universality can be achieved when the output layer is a convolutional filter of width 1, without the requirement of sufficient depth as in Theorem 2 and Corollary 1. This is formalized as follows.

**Theorem 3.** *Let $V_0, \ldots, V_f$ be permutation representations of a finite group $G$. Let $X$ be a finite $G$-set and $\mathbb{R}^X$ its associated permutation representation. Let $M_1, \ldots, M_f$ be layer spaces in $\mathrm{Aff}_G(V_{i-1}, V_i)$ for $i = 1, \ldots, f$, and let $C$ be a layer space in $\mathrm{Aff}_G(\mathbb{R}^X, \mathbb{R}^X)$ of convolutional filters with width 1 as defined in Example 1.iii. Then $\mathcal{U}_\sigma(M_1, \ldots, M_f, C) = \mathcal{C}_{\overline{\rho}}(V)$, where $\overline{\rho} := \overline{\rho}\big(\mathcal{U}_\sigma(M_1, \ldots, M_f, C)\big).$*

Note that when $C$ is defined on a one-dimensional space, we have $C = L$, and $M^d$ becomes the invariant layer space $I$. In this case, Theorem 3, which is formulated in the equivariant setting, specializes to Theorem 1—the corresponding result in the invariant setting.

At first sight, it may be tempting to compare Theorem 2 and Theorem 3 and conclude that Theorem 3 is a stronger statement. However, it is important to note that adding the $C$ layer space at the end does not change the entry-wise separation power of the model class, whereas adding a certain number of $M$ layers may increase it. Theorem 2 explicitly accounts for this effect.

Theorem 2 and Corollary 1 may be particularly relevant for their practical implications: they ensure that maximal expressivity is reached at finite depth and rule out the possibility of unbounded improvement. Theorem 3, on the other hand, is instrumental in recovering known results such as (Segol & Lipman, 2020). It also shows that universality stabilization in Theorem 2 and Corollary 1 can occur at the same depth as entry-wise separation stabilization, revealing that the threshold in Theorem 2 is not always optimal.

*Remark* 1. Thanks to Theorem 3, we can easily recover the universality result of Segol & Lipman (2020). Namely, $\mathcal{U}_\sigma(C, P, C) = \mathcal{U}_\sigma(P, P, P) = \mathcal{C}_{S_n}(\mathbb{R}^n, \mathbb{R}^n)$. It remains to verify that $\pi_i^* \mathcal{N}_\sigma(C, P, C)$ separates $\mathrm{Stab}_{S_n}(i)$-orbits in $\mathbb{R}^n$, which follows directly from Lemma 6 (Appendix C.2).

Note that this shows that the depth threshold required for separation-stability in Theorem 2 provides a *sufficient*, but not *necessary*, condition for universality. Indeed, $\overline{\rho}\big(\mathcal{U}_\sigma(V, \mathbb{R}^G, \mathbb{R}^G, W)\big) = \overline{\rho}\big(\mathcal{U}_\sigma(V, \mathbb{R}^G, W)\big)$, so separation has not stabilized, yet entry-wise separation universality is already achieved. However, determining in general when separation stabilization takes place is a difficult problem. Corollary 1 guarantees that maximal expressivity is reached after a finite number of steps and then saturates. This result supports the intuition that increasing depth enhances expressivity. Less

intuitively, it also shows that beyond a certain threshold, saturation occurs and further increases in depth no longer affect the universality class.

*Remark* 2. Theorem 3 marks a significant difference between the equivariant and the invariant cases. Indeed, Pacini et al. (2025b) shows that, although $\rho(\mathcal{U}_\sigma(C, I)) = \rho(\mathcal{U}_\sigma(P, I)) = \rho(\mathcal{C}_{S_n}(\mathbb{R}^n, \mathbb{R}))$, the corresponding universality classes satisfy $\mathcal{U}_\sigma(C, I) \subsetneq \mathcal{U}_\sigma(P, I) \subsetneq \mathcal{C}_{S_n}(\mathbb{R}^n, \mathbb{R})$. These strict inequalities are proved via a characterization through differential operators. In the equivariant case, we have $\mathcal{U}_\sigma(C, C) \subsetneq \mathcal{U}_\sigma(P, C) \subseteq \mathcal{C}_{S_n}(\mathbb{R}^n, \mathbb{R}^n)$, yet as we showed here, both spaces can be characterized in terms of entry-wise separation, without resorting to the differential operator characterization. Note that we expect this to be a phenomenon specific to networks with output layers in $\mathrm{Aff}_G(\mathbb{R}^X, \mathbb{R}^X)$. Output spaces in $\mathrm{Aff}_G(\mathbb{R}^X, \mathbb{R})$, or more generally in $\mathrm{Aff}_G(\mathbb{R}^X, \mathbb{R}^Y)$, may instead require a characterization in terms of differential operators for arbitrary finite $G$-sets $Y$.

## 6  LIMITATIONS

Our results characterize universality of deep invariant and equivariant networks under separation constraints, but several limitations remain which provide avenues for future work. First, the theory applies to networks with point-wise activations and permutation representations. Extending the analysis to other types of representations or more general nonlinearities may require different approaches. Second, our universality theorems are asymptotic and do not provide quantitative approximation rates or sample complexity bounds, which are important for understanding expressivity in practice. Finally, we have not addressed optimization or trainability: while depth is shown to be sufficient for universality, when such networks can be efficiently trained remains an open question.

## 7  CONCLUSIONS

We established new universality results for deep invariant and equivariant networks. For invariance, we proved that depth is sufficient to guarantee universality within the class of separation-respecting functions. For equivariance, we introduced the refined concept of entry-wise separability and showed that, once entry-wise relations stabilize, deep equivariant networks achieve universality. Taken together, these results unify and extend prior shallow or architecture-specific universality theorems, highlighting depth as a general mechanism for universality in equivariant models. We hope this framework provides a basis for future advances in the analysis and design of expressive, symmetry-aware, neural networks.

## ACKNOWLEDGMENTS

This work was carried out while Marco Pacini was visiting the Geometric Learning Lab at Northeastern University. The work of Mircea Petrache was supported by National Center for Artificial Intelligence CENIA FB210017, Basal ANID. The work of Bruno Lepri was partially supported by the following projects: Horizon Europe Programme, grants #10112- 0237-ELIAS and #101120763-TANGO. Funded by the European Union. Views and opinions expressed are however those of the author(s) only and do not necessarily reflect those of the European Union or the European Health and Digital Executive Agency (HaDEA). Neither the European Union nor the granting authority can be held responsible for them. This work was also partly supported by Ministero delle Imprese e del Made in Italy (IPCEI Cloud DM 27 giugno 2022 – IPCEI-CL-0000007) and European Union (Next Generation EU).

## REPRODUCIBILITY STATEMENT

This work is purely theoretical and contains no experiments or datasets. All results are formally stated as theorems, propositions, or corollaries, and complete proofs are provided in the main text and appendices. Definitions and assumptions are explicitly stated to ensure mathematical clarity, and we reference relevant prior results where appropriate. As such, all claims in the paper can be fully verified by checking the provided proofs.

## ETHICS STATEMENT

This work is theoretical and does not involve experiments with human subjects, sensitive data, or deployment of models in real-world applications. We therefore do not foresee any direct ethical concern.

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

## A PRELIMINARIES

**Definition 7** (Group). *A group is a set $G$ together with a binary operation $\cdot : G \times G \to G$ such that:*

- *Associativity: for all $g, h, k \in G$ we have $(g \cdot h) \cdot k = g \cdot (h \cdot k)$.*

- *Identity element: there exists an element $e \in G$ such that $g \cdot e = e \cdot g = g$ for every $g \in G$.*

- *Inverses: for every $g \in G$ there exists an element $g^{-1} \in G$ such that $g \cdot g^{-1} = g^{-1} \cdot g = e$.*

*The group is* finite *if $G$ has finitely many elements.*

We next recall the notion of a group homomorphism, which is a structure-preserving map between groups.

**Definition 8** (Homomorphism). *Let $G$ and $H$ be groups. A function $\phi : G \to H$ is called a* group homomorphism *if, for all $g, h \in G$, it satisfies*

$$\phi(g \cdot h) = \phi(g) \cdot \phi(h).$$

**Definition 9** (Group Actions). *Let $G$ be a group and let $X$ be a set. A* group action *of $G$ on $X$ is a map*

$$\Phi : G \times X \to X,$$

*often written as $\phi_g(x) = \Phi(g, x)$ for $g \in G$ and $x \in X$, such that:*

- *Identity: $\phi_e = \mathrm{id}_X$, where $e$ is the identity element of $G$.*

- *Compatibility: for all $g, h \in G$ we have $\phi_g \circ \phi_h = \phi_{gh}$.*

*In practice, we frequently denote the action by $g \cdot x$ or simply $gx$ instead of $\phi_g(x)$.*

*A set $X$ together with a group action of $G$ is called a $G$-set. Equivalently, $X$ is a $G$-set if there exists an action $\cdot : G \times X \to X$ satisfying the identity and compatibility conditions above.*

A central notion in our analysis is that of a map between $G$-sets that respects the group action, leading to the following definition.

**Definition 10** (Equivariance). *Let $X$ and $Y$ be $G$-sets. A function $f : X \to Y$ is $G$-equivariant if, for all $g \in G$ and $x \in X$, we have*

$$f(g \cdot x) = g \cdot f(x).$$

**Definition 11** (Group Representations). *Let $G$ be a group and let $V$ be a vector space over $\mathbb{R}$. A* representation *of $G$ on $V$ is a group homomorphism*

$$\phi : G \to \mathrm{GL}(V),$$

*where $\mathrm{GL}(V)$ denotes the group of invertible linear maps $V \to V$. Given such a homomorphism, we obtain a linear action of $G$ on $V$ by setting*

$$gv := \phi(g)(v) \quad \text{for } g \in G, \ v \in V.$$

When $V$ and $W$ are $G$-representations, we denote by $\mathrm{Hom}_G(V, W)$ the space of $G$-equivariant linear maps $V \to W$, and by $\mathrm{Aff}_G(V, W)$ the set of $G$-equivariant affine maps $V \to W$.

Note that $\mathrm{Hom}_G(V, W)$ is a vector space. Indeed, $0 \in \mathrm{Hom}_G(V, W)$, and for any $f, g \in \mathrm{Hom}_G(V, W)$ and $\alpha, \beta \in \mathbb{R}$, the linear combination $\alpha f + \beta g$ is still in $\mathrm{Hom}_G(V, W)$. The same property holds for $\mathrm{Aff}_G(V, W)$.

In this paper, we mainly work with permutation representations. This is not merely a mild restriction when considering point-wise activations: as shown in Pacini et al. (2024), the only representations of compact groups that are compatible with all continuous activation functions are permutation representations.

**Definition 12** (Permutation Representations). *Let $X$ be a finite set and let $G$ be a finite group acting on $X$. The associated* permutation representation *of $G$ is the linear action of $G$ on $\mathbb{R}^X$ defined on the standard basis $\{e_x\}_{x \in X}$ by*

$$g(e_x) = e_{g \cdot x} \quad \text{for all } g \in G, \; x \in X.$$

We now make explicit how this fits the general notion of a representation.

**Proposition 3.** *Let $X$ and $G$ be as above and set $V := \mathbb{R}^X$. For each $g \in G$ there is a unique linear map*

$$\phi(g) : V \to V$$

*such that $\phi(g)(e_x) = e_{g \cdot x}$ for all $x \in X$. Then the map*

$$\phi : G \to \mathrm{GL}(V), \quad g \mapsto \phi(g)$$

*is a representation of $G$ on $V$.*

*Proof.* Since $\{e_x\}_{x \in X}$ is a basis of $V = \mathbb{R}^X$, for each $g \in G$ there exists a unique linear map $\phi(g) : V \to V$ such that $\phi(g)(e_x) = e_{g \cdot x}$ for all $x \in X$. Moreover,

$$\phi(g^{-1})(\phi(g)(e_x)) = \phi(g^{-1})(e_{g \cdot x}) = e_{g^{-1} \cdot (g \cdot x)} = e_x,$$

so $\phi(g^{-1})$ is the inverse of $\phi(g)$ and $\phi(g) \in \mathrm{GL}(V)$.

For $g, h \in G$ and any $x \in X$ we have

$$(\phi(g)\phi(h))(e_x) = \phi(g)(e_{h \cdot x}) = e_{g \cdot (h \cdot x)} = e_{(gh) \cdot x} = \phi(gh)(e_x),$$

hence $\phi(g)\phi(h) = \phi(gh)$ and $\phi : G \to \mathrm{GL}(V)$ is a group homomorphism. $\qquad \square$

Moreover, after choosing an ordering $X = \{x_1, \ldots, x_n\}$ and identifying $V \cong \mathbb{R}^n$, each $\phi(g)$ is represented by a permutation matrix $P_g \in \mathbb{R}^{n \times n}$ with entries

$$(P_g)_{ij} = 1 \text{ if } g \cdot x_j = x_i, \text{ and } (P_g)_{ij} = 0 \text{ otherwise.}$$

In particular, $P_{gh} = P_g P_h$ and $P_e = I_n$, so $g \mapsto P_g$ is a group homomorphism into $\mathrm{GL}_n(\mathbb{R})$.

## B  SEPARATION-CONSTRAINED UNIVERSALITY FOR INVARIANT NETWORKS

We recall the notation introduced in equation 3 for the subspace $M$ of $\mathrm{Aff}_G(\mathbb{R}^X, \mathbb{R}^Y)$.

$$M^{k \times h} := \left\{ \begin{array}{c} (x_1, \ldots, x_k) \mapsto \left( \sum_{j=1}^k f_{1,j}(x_j), \ldots, \sum_{j=1}^k f_{h,j}(x_j) \right) \\ f_{ij} \in M, i = 1, \ldots, h, j = 1, \ldots, k \end{array} \right\}.$$

Thus, $M^{k \times h} \subseteq \mathrm{Aff}_G(\mathbb{R}^X \otimes \mathbb{R}^k, \mathbb{R}^Y \otimes \mathbb{R}^h)$. In particular, for each $f_{ij}$ in the above definition, we write

$$f_{ij}(x) = A_{ij}x + b_{ij},$$

where $A_{ij}$ and $b_{ij}$ denote respectively the linear part and the translational part of $f_{ij}$. With this notation, the linear and translational parts of an element in $M^{k \times h}$ can be written respectively as

$$\begin{bmatrix} A_{11} & A_{12} & \cdots & A_{1k} \\ A_{21} & A_{22} & \cdots & A_{2k} \\ \vdots & \vdots & \ddots & \vdots \\ A_{h1} & A_{h2} & \cdots & A_{hk} \end{bmatrix} \quad \text{and} \quad \begin{bmatrix} \sum_{j=1}^k b_{1j} \\ \sum_{j=1}^k b_{2j} \\ \vdots \\ \sum_{j=1}^k b_{hj} \end{bmatrix}.$$

The following definitions and lemmas are used for the proof of Theorem 1.

**Lemma 1.** *Let* $f_1, \ldots, f_h \in \mathcal{N}_\sigma(M_1, \ldots, M_d, I)$ *then their parallelization* $(f_1, \ldots, f_h)$ *belongs to* $\bigcup_{\tilde{k} \in \mathbb{N}^{d+1}} \mathcal{N}_\sigma(M_1^{1 \times k_1}, \ldots, M_d^{k_{d-1} \times h})$.

*Proof of Lemma 1.* Let us consider the case $h = 2$; for $h > 2$, the proof is analogous.

Let $f_1, f_2 \in \mathcal{N}_\sigma(M_1, \ldots, M_d)$. Then each affine layer at depth $i$ in $(f_1, f_2)$ is a block diagonal matrix whose first block is the $i$-th layer of $f_1$ and the second is the $i$-th layer of $f_2$; a similar analysis holds for the bias terms. Hence, this layer belongs to $M_i^{2 \times 2}$ for $i > 1$. For $i = 1$, the first linear layer of $(f_1, f_2)$ is a block column matrix where each block is the first layer of $f_1$ and $f_2$; again, a similar analysis holds for the bias terms. Hence, this layer belongs to $M_1^{1 \times 2}$. This shows that

$$(f_1, f_2) \in \mathcal{N}_\sigma(M_1^{1 \times 2}, M_2^{2 \times 2}, \ldots, M_d^{2 \times 2}) \subseteq \bigcup_{\tilde{k} \in \mathbb{N}^{d-1}} \mathcal{N}_\sigma(M_1^{1 \times k_1}, \ldots, M_d^{k_{d-1} \times 2}).$$

$\square$

**Definition 13.** *Let* $M_1, \ldots, M_d$ *be layer spaces. Let* $\mathcal{B}_i$ *be bases for the layer space* $M_i$, *and define*

$$M_i^\mathbb{Q} := \operatorname{Span}_\mathbb{Q} \mathcal{B}_i$$

*for each* $i = 1, \ldots, d$. *Define rational neural spaces as follows:*

$$\mathcal{N}_\sigma{}^\mathbb{Q}(M_1, \ldots, M_d) := \mathcal{N}_\sigma(M_1^\mathbb{Q}, \ldots, M_d^\mathbb{Q}).$$

*Note that* $\mathcal{N}_\sigma{}^\mathbb{Q}(M_1, \ldots, M_d)$ *depends on the choice of the bases* $\mathcal{B}_1, \ldots, \mathcal{B}_d$.

**Lemma 2.** *In the notation of Definition 13,*

$$\rho(\mathcal{N}_\sigma(M_1, \ldots, M_d)) = \rho(\mathcal{N}_\sigma{}^\mathbb{Q}(M_1, \ldots, M_d)).$$

*Therefore,* $\rho(\mathcal{N}_\sigma{}^\mathbb{Q}(M_1, \ldots, M_d))$ *does not depend on the choice of bases* $\mathcal{B}_1, \ldots, \mathcal{B}_d$.

*Proof of Lemma 2.* By the continuity of the parametrization map and the density of $M_i^\mathbb{Q}$ in $M_i$. $\square$

Lemma 2 implies the following corollary.

**Corollary 2.** *There exists a countable family* $\mathcal{F} = \{f_h\}_{h \in \mathbb{N}} \subseteq \mathcal{N}_\sigma(M_1, \ldots, M_d)$ *such that*

$$\rho(\mathcal{F}) = \rho(\mathcal{N}_\sigma(M_1, \ldots, M_d)).$$

**Lemma 3.** *Let* $V = \mathbb{R}^d$ *with its usual topology and let* $\rho$ *be a* closed *equivalence relation on* $V$. *For a family* $\mathcal{F} = \{f_n\}_{n \in \mathbb{N}}$ *of continuous maps* $f_n : V \to \mathbb{R}^m$ *such that* $\rho(\mathcal{F}) = \rho$. *Then the set*

$$\mathcal{A} := \bigcup_{n \geq 1} \left\{ A(f_1, \ldots, f_n)|_K \ : \ A \in \mathcal{C}\left((\mathbb{R}^m)^n, \mathbb{R}\right) \right\}$$

*is dense in* $\mathcal{C}_\rho(V)$. *Or equivalently, for every* $h \in \mathcal{C}_\rho(V)$ *there exist* $n_k \uparrow \infty$ *and* $A_{n_k} \in \mathcal{C}((\mathbb{R}^m)^{n_k}, \mathbb{R})$ *such that* $A_{n_k}(f_1, \ldots, f_{n_k}) \to h$.

*Proof of Lemma 3.* For $x \in V$ set $\widehat{F}(x) := (f_n(x))_{n \in \mathbb{N}}$. Fix a compact $K \subset V$. Since each $f_n$ is continuous, $\widehat{F}(K)$ is compact in the product $V^\mathbb{N}$. Note that $\rho = \{(x, y) \in V^2 : \widehat{F}(x) = \widehat{F}(y)\}$, so that the map $\phi : K/\rho \to \widehat{F}(K)$ defined by $\phi([x]) := \widehat{F}(x)$ is well defined. Furthermore $\phi$ is continuous and bijective, and since $K/\rho$ is compact Hausdorff and $\widehat{F}(K)$ is Hausdorff, $\phi$ is a homeomorphism. Hence every $h \in \mathcal{C}_\rho(K)$ factors uniquely as

$$h = H \circ \widehat{F}|_K \qquad \text{for a unique } H \in \mathcal{C}(\widehat{F}(K), \mathbb{R}).$$

Let $\pi_n : (\mathbb{R}^m)^\mathbb{N} \to (\mathbb{R}^m)^n$ be the projection onto the first $n$ coordinates. Note that

$$\mathcal{A} = \bigcup_{n \geq 1} \left\{ A \circ \pi_n|_{\widehat{F}(K)} \ : \ A \in \mathcal{C}((\mathbb{R}^m)^n, \mathbb{R}) \right\}.$$

Then $\mathcal{A}$ is a sub-algebra of $\mathcal{C}(\widehat{F}(K))$ containing constants. We next prove that $\mathcal{A}$ separates points. Indeed, if $y, y' \in \widehat{F}(K)$ with $y \neq y'$, then there exists $j$ with $y_j \neq y'_j$; define the continuous scalar function $p : \mathbb{R}^m \to \mathbb{R}$ such as $p(u) = \langle u, y_j - y'_j \rangle$, and note that $p(y_j) \neq p(y'_j)$ and choose $A \in \mathcal{C}((\mathbb{R}^m)^n)$ given by $A(z_1, \ldots, z_j, \ldots) := p(z_j)$. This function $A$ lies in $\mathcal{A}$ and satisfies $(A \circ \pi_j)(y) \neq (A \circ \pi_j)(y')$ as desired. Now we may use the Stone–Weierstrass theorem, which gives that $\overline{\mathcal{A}} = \mathcal{C}(\widehat{F}(K))$ in the uniform norm, concluding the proof. $\qquad\square$

## C  UNIVERSALITY OF EQUIVARIANT NEURAL NETWORKS

### C.1  ENTRY-WISE SEPARATION

In this section, we study equivariant functions by reducing the problem to the analysis of particular invariant functions, thereby extending the previous results. The tools used for this reduction are suitable projections onto output coordinates, together with reconstruction maps that allow us to recover the entire function from a single projection. For the sake of presentation, we begin by considering the case where $G$ acts transitively on $X$. Let $x \in X$, and let $G_x$ denote the stabilizer of $x$. Let $\pi_x : \mathbb{R}^X \to \mathbb{R}$ be the linear projection on the $x$-th coordinate in $\mathbb{R}^X$. We obtain the pushforward map of $\pi_x$, defined as

$$\pi_{x*} : \quad \begin{array}{c} \mathcal{C}_G(V, \mathbb{R}^X) \to \mathcal{C}_{G_x}(V) \\ f \mapsto \pi_x \circ f. \end{array}$$

Let $g_1, \ldots, g_t$ be a transversal for $G/G_x$, that is, a choice of representatives for classes in $G/G_x$. We define the *reconstruction map* as

$$\theta_x^* : \quad \begin{array}{c} \mathcal{C}_{G_x}(V) \to \mathcal{C}_G(V, \mathbb{R}^X) \\ f \mapsto \left[ v \mapsto \sum_{i=1}^t f(g_i^{-1} v) e_{g_i x} \right]. \end{array}$$

**Proposition 4.** *If the action of $G$ on $X$ is transitive, then reconstruction map $\theta_x^*$ is a well-defined, continuous linear operator such that*

*(i)* $\pi_{x*} \circ \theta_x^* = \mathrm{id}_{\mathcal{C}_{G_x}(V)}$,

*(ii)* $\theta_x^* \circ \pi_{x*} = \mathrm{id}_{\mathcal{C}_G(V, \mathbb{R}^X)}$.

*Proof.* Choosing a different representative for each $g_i$ means choosing an element $g_i \cdot h$ for an arbitrary $h \in G_x$. For $f \in \mathcal{C}_{G_x}(V)$, the $G_x$-invariance of $f$ implies

$$f((g_i \cdot h)^{-1} v) = f(h^{-1} \cdot g_i^{-1} v) = f(g_i^{-1} v).$$

Then $e_{g_i h x} = e_{g_i x}$ since $h \in G_x$. As a consequence, the choice of representatives $g_1, \ldots, g_t$ does not affect $\theta_x^*(f)$. Next, we prove that $\theta_x^*(f)$ is $G$-equivariant: indeed, for $g \in G$ we have

$$\begin{aligned}
\theta_x^*(f)(gv) &= \sum_{i=1}^t f\left(g_i^{-1} gv\right) e_{g_i x} \\
&= \sum_{i=1}^t f\left(g_i^{-1} v\right) e_{g^{-1} g_i x} \\
&= g \cdot \sum_{i=1}^t f\left(g_i^{-1} v\right) e_{g_i x} \\
&= g \cdot \theta_x^*(f)(v).
\end{aligned}$$

where in the second equality we use the fact that $g^{-1} g_1, \ldots, g^{-1} g_t$ is another transversal for $G/G_x$. These observations prove that $\theta_x^*$ is well-defined. It is continuous and linear since it is the composition of continuous and linear functions. We can choose $g_1 = e$, in which case the $x$-th coefficient in $\theta_x^*(f)(v)$ is simply $f(v)$, proving (i). To prove (ii), notice that for $x \in X$, the set $g_1, \ldots, g_t$ is a

transversal of $G/G_x$ if and only if $g_1 x, \ldots, g_t x$ is the $G$-orbit of $x$. Thus for each $f \in \mathcal{C}(V, \mathbb{R}^X)$ we can write

$$f(v) = \sum_{i=1}^{t} \pi_{g_i x} f(v) e_{g_i x}. \tag{9}$$

Now for $f \in \mathcal{C}_G(V, \mathbb{R}^X)$, we can conclude (ii) as follows:

$$\theta_x^* \pi_{x*} f(v) = \sum_{i=1}^{t} \pi_x f(g_i^{-1} v) \, e_{g_i x}$$

$$= \sum_{i=1}^{t} \pi_x \big( g_i^{-1} \cdot f(v) \big) \, e_{g_i x}$$

$$= \sum_{i=1}^{t} \pi_{g_i x} f(v) \, e_{g_i x}$$

$$\overset{\text{Equation 9}}{=} f(v).$$

$\square$

Proposition 4 says that $\pi_{x*}$ is a linear homeomorphism, hence, a function class $\mathcal{N}$ is dense in $\mathcal{C}_G(V, \mathbb{R}^X)$ if and only if $\pi_{x*}(\mathcal{N})$ is. This means that we can restrict ourselves to the study of function families of type $\pi_{x*}(\mathcal{N})$, which are similar to the study conducted in Section 3.3.

*Proof of Proposition 1.* The claim follows directly from Proposition 4: applying $\pi_x$ yields one inclusion, while the reconstruction map yields the other. $\square$

*Remark* 3 (Linear case). In particular, in the affine case we obtain,

$$\pi_{x*} : \quad \begin{array}{c} \mathrm{Aff}_G(V, \mathbb{R}^X) \to \mathrm{Aff}_{G_x}(V, \mathbb{R}) \\ f \mapsto \pi_x \circ f. \end{array}$$

Note that characterizing $\mathrm{Aff}_{G_x}(V, \mathbb{R})$ reduces to computing $V^{G_x}$. If $V = \mathbb{R}^Y$ for a finite $G$-set $Y$, then we just need to compute the orbits of $G_x$ on $Y$.

The previous observations extend, with minor modifications, to the non-transitive case, which we address next. Let $X = Y_1 \sqcup \cdots \sqcup Y_s$ be the decomposition of $X$ into $G$-orbits. For each $i = 1, \ldots, s$, denote by $\pi_{\mathbb{R}^{Y_i}} : \mathbb{R}^X \to \mathbb{R}^{Y_i}$ the standard projection. Consider the maps

$$\Phi : \quad \begin{array}{c} \mathcal{C}_G(V, \mathbb{R}^X) \to \mathcal{C}_G(V, \mathbb{R}^{Y_1}) \oplus \cdots \oplus \mathcal{C}_G(V, \mathbb{R}^{Y_s}) \\ f \mapsto (\pi_{\mathbb{R}^{Y_1}} f, \ldots, \pi_{\mathbb{R}^{Y_s}} f), \end{array}$$

and

$$\Psi : \quad \begin{array}{c} \mathcal{C}_G(V, \mathbb{R}^{Y_1}) \oplus \cdots \oplus \mathcal{C}_G(V, \mathbb{R}^{Y_s}) \to \mathcal{C}_G(V, \mathbb{R}^X) \\ (f_1, \ldots, f_s) \mapsto \big[ x \mapsto f_1(x) + \cdots + f_s(x) \big]. \end{array}$$

Note that $\Phi$ is a homeomorphism onto its image. Let $x_1, \ldots, x_s$ be points with $x_i \in Y_i$. Then, for each $i = 1, \ldots, s$, we have

$$\pi_{x_i^*} \, \mathcal{C}_G(V, \mathbb{R}^X) = \pi_{x_i^*} \, \mathcal{C}_G(V, \mathbb{R}^{Y_i}).$$

Moreover, for each $i = 1, \ldots, s$,

$$\mathcal{C}_G(V, \mathbb{R}^{Y_i}) = \theta_{x_i}^* \, \pi_{x_i^*} \, \mathcal{C}_G(V, \mathbb{R}^{Y_i}) = \mathcal{C}_{G_{x_i}}(V, \mathbb{R}).$$

Therefore, understanding $\mathcal{C}_G(V, \mathbb{R}^X)$ reduces to understanding $\mathcal{C}_{G_{x_i}}(V, \mathbb{R})$ for each $i = 1, \ldots, s$, and then reconstructing $\mathcal{C}_G(V, \mathbb{R}^X)$ via $\theta_{x_1}^*, \ldots, \theta_{x_s}^*$ and $\Psi$. In particular, since we focus on closed linear subspaces $\mathcal{U} \subseteq \mathcal{C}_G(V, \mathbb{R}^X)$, it is enough to study the subspaces $\pi_{x_i^*} \mathcal{U}$ for $i = 1, \ldots, s$, namely universality classes of invariant neural networks.

**Proposition 5.** *The following equality is true:*

$$\mathcal{U}_\sigma(\underbrace{C, \ldots, C}_{d \text{ times}}) = \{ (x_1, \ldots, x_n) \mapsto (f(x_1), \ldots, f(x_n)) \mid f \in \mathcal{C}(\mathbb{R}) \} \subsetneq \mathcal{C}_{S_n}(\mathbb{R}^n, \mathbb{R}^n). \tag{10}$$

*Proof of Proposition 5.* We start by considering the case $d = 2$ and then study the more general neural space $\mathcal{N}_\sigma(C^{1,h}, C^{h \times k})$.

Recall $\lambda(C) = \text{Span}\{x \mapsto id_{\mathbb{R}^X} \cdot x\}$. Elements in $C^{1,h}$ can be represented as affine maps $x \mapsto Bx + c$ where $B$ and $c$ have the following block representations

$$B = \begin{bmatrix} b_1 \, \text{id} \\ \vdots \\ b_h \, \text{id} \end{bmatrix} \quad \text{and} \quad c = \begin{bmatrix} c_1 \, \mathbb{1} \\ \vdots \\ c_h \, \mathbb{1} \end{bmatrix}.$$

While elements in $C^{h,k}$ can be represented as affine maps $x \mapsto Ax + d$ where $d \in \mathbb{R}$ and

$$A = \begin{bmatrix} a_{1,1} \cdot id_{\mathbb{R}^X} & \cdots & a_{1,h} \cdot id_{\mathbb{R}^X} \\ \vdots & \vdots & \vdots \\ a_{k,1} \cdot id_{\mathbb{R}^X} & \cdots & a_{h,h} \cdot id_{\mathbb{R}^X} \end{bmatrix} = \tilde{A} \otimes id_{\mathbb{R}^X},$$

where $\tilde{A} = [a_{i,j}] \in \mathbb{R}^{k \times h}$.

Given $i \in X$ and $s = 1, \ldots, h$, we can write elements $\theta \in \mathcal{N}_\sigma(C^{1,h}, C^{h,k})$ as

$$\theta_{s,i}(x) = A\sigma(Bx + c) = \sum_{j=1}^{h} a_{s,j} \sigma\left(b_j x_i + c_j\right)$$

for some $a_i, b_j, c_j \in \mathbb{R}$. But note that

$$\theta_{s,i}(x) = \sum_{j=1}^{h} a_{s,j} \sigma\left(b_j x_i + c_j\right) = \xi_s(x) \tag{11}$$

where

$$\xi_s(y) := \sum_{j=1}^{h} a_{s,j} \sigma\left(b_j y + c_j\right).$$

That is, $\xi \in \mathcal{N}_\sigma(\mathbb{R}^m, \mathbb{R}^h, \mathbb{R}^k)$. In other words, taking the limit as $h \to \infty$ and setting $k = 1$, we obtain the proof of the theorem for the case $d = 2$. For $d > 2$, it suffices to note that the composition of spaces of the type $\mathcal{N}_\sigma(C^{1,h}, C^{h \times k})$ again yields elements of the same type. This concludes the proof. $\qquad \square$

## C.2 Entry-wise Separation Constrained Universality

*Proof of Theorem 3.* Recall that
$$C \subseteq \text{Aff}_G(\mathbb{R}^X, \mathbb{R}^X).$$

Thanks to Proposition 4, to prove Theorem 3 it suffices to show that, for each $x \in X$,
$$\pi_{x*} \, \mathcal{U}_\sigma(M_1, \ldots, M_f, C) = \mathcal{C}_{\rho_x}(V),$$

where
$$\rho_x := \rho\left(\pi_{x*} \, \mathcal{U}_\sigma(M_1, \ldots, M_f, C)\right).$$

Fix $x \in X$, and define $P_x := \pi_{x*}C$. Then

$$\begin{aligned} \pi_{x*} \, \mathcal{U}_\sigma(M_1, \ldots, M_f, C) &= \pi_{x*} \, \mathcal{U}_\sigma(M_1, \ldots, M_f, \pi_{x*}C) \\ &= \mathcal{U}_\sigma(M_1, \ldots, M_f, P_x) \\ &= \mathcal{U}_\sigma(M_1, \ldots, M_{f-1}, \pi_{x*}M_f, L). \end{aligned}$$

For the second equality, note that

$$\begin{aligned} P_x &= \pi_x^* C \\ &= \{v \mapsto \pi_x(\lambda v + \mu \mathbb{1}) \mid \lambda, \mu \in \mathbb{R}\} \\ &= \{v \mapsto \lambda \pi_x(v) + \mu \mid \lambda, \mu \in \mathbb{R}\}. \end{aligned}$$

For the third equality, recall that the pointwise activation $\tilde{\sigma}$ is defined by

$$\tilde{\sigma}\Big( \sum_{x \in X} v_x e_x \Big) := \sum_{x \in X} \sigma(v_x)\, e_x,$$

and observe that, for each $x \in X$, we have the commutation relation

$$\sigma \circ \pi_x = \pi_x \circ \tilde{\sigma}.$$

Hence, at the final activation we have

$$\{\, v \mapsto \lambda \pi_x(\tilde{\sigma}(v)) + \mu \mid \lambda, \mu \in \mathbb{R} \,\} = \{\, v \mapsto \lambda \sigma(\pi_x(v)) + \mu \mid \lambda, \mu \in \mathbb{R} \,\},$$

and note that, in this way, the final layer becomes the space of arbitrary affine maps of the real line, namely $L$. Finally, note that $M_f \subseteq \mathrm{Aff}_G(V, \mathbb{R}^X)$ for some permutation representation $V$. Thus, by Remark 3, we have

$$\pi_{x*} M_f \subseteq \mathrm{Aff}_{G_x}(V, \mathbb{R}),$$

and therefore $\pi_{x*} M_f$ is a space of $G_x$-invariant affine functions. Thanks to Theorem 1, we obtain

$$\pi_{x*}\, \mathcal{U}_\sigma(M_1, \ldots, M_f, C) = \mathcal{U}_\sigma(M_1, \ldots, M_{f-1}, \pi_{x*} M_f, L) = \mathcal{C}_{\rho_x}(V),$$

where

$$\rho_x = \rho\big(\pi_{x*}\, \mathcal{U}_\sigma(M_1, \ldots, M_f, C)\big) = \rho\big(\mathcal{U}_\sigma(M_1, \ldots, M_{f-1}, \pi_{x*} M_f, L)\big).$$

This concludes the proof. $\square$

For brevity, we will prove all subsequent results in terms of $\mathcal{U}_\sigma(\underbrace{M, ..., M}_{d \text{ times}})$ or similar forms. The same results, however, extend verbatim to the more general setting $\mathcal{U}_\sigma(M_1, \ldots, M_d, \underbrace{M, ..., M}_{d \text{ times}})$.

*Proof of Theorem 2.* Note that

$$\mathcal{U}_\sigma(\underbrace{M, \ldots, M}_{d \text{ times}}, C) \subseteq \mathcal{U}_\sigma(\underbrace{M, \ldots, M}_{d+1 \text{ times}}).$$

Therefore,

$$\rho\big(\mathcal{U}_\sigma(\underbrace{M, \ldots, M}_{d+1 \text{ times}})\big) \subseteq \rho\big(\mathcal{U}_\sigma(\underbrace{M, \ldots, M}_{d \text{ times}}, C)\big) \subseteq \rho\big(\mathcal{U}_\sigma(\underbrace{M, \ldots, M}_{d \text{ times}})\big).$$

By hypothesis,

$$\rho := \rho\big(\mathcal{U}_\sigma(\underbrace{M, \ldots, M}_{d+1 \text{ times}})\big) = \rho\big(\mathcal{U}_\sigma(\underbrace{M, \ldots, M}_{d \text{ times}})\big),$$

and hence

$$\rho = \rho\big(\mathcal{U}_\sigma(\underbrace{M, \ldots, M}_{d \text{ times}}, C)\big)$$

as well. Then, by Theorem 3,

$$\mathcal{U}_\sigma(\underbrace{M, \ldots, M}_{d \text{ times}}, C) = \mathcal{C}_\rho(V).$$

Since functions realized by neural networks are continuous, it follows that

$$\mathcal{U}_\sigma(\underbrace{M, \ldots, M}_{d+1 \text{ times}}) \subseteq \mathcal{C}_\rho(V).$$

Finally, we observe that

$$\mathcal{C}_\rho(V) = \mathcal{U}_\sigma(\underbrace{M, \ldots, M}_{d \text{ times}}, C) \subseteq \mathcal{U}_\sigma(\underbrace{M, \ldots, M}_{d+1 \text{ times}}) \subseteq \mathcal{C}_\rho(V),$$

which yields the claim. $\square$

**Lemma 4.** *Let $X, Y, Z$ be metric spaces, and let $\mathcal{G} \subseteq \mathcal{C}(X, Y)$ and $\mathcal{F} \subseteq \mathcal{C}(Y, Z)$. The following identities hold.*

$$\rho(\overline{\mathcal{G}} \mathbin{\hat{\circ}} \mathcal{F}) = \rho(\mathcal{G} \mathbin{\hat{\circ}} \mathcal{F}) = \rho(\mathcal{G} \mathbin{\hat{\circ}} \overline{\mathcal{F}}),$$

*where the closure is taken with respect to uniform convergence on compact sets.*

*Proof.* For convenience, recall the definition

$$\mathcal{G} \mathbin{\hat{\circ}} \mathcal{F} := \{f \circ g \mid f \in \mathcal{F}, g \in \mathcal{G}\}.$$

We start by proving the first equality. Since $\mathcal{G} \mathbin{\hat{\circ}} \mathcal{F} \subseteq \overline{\mathcal{G}} \mathbin{\hat{\circ}} \mathcal{F}$, we have

$$\rho(\overline{\mathcal{G}} \mathbin{\hat{\circ}} \mathcal{F}) \subseteq \rho(\mathcal{G} \mathbin{\hat{\circ}} \mathcal{F}).$$

For the reverse inclusion, let $(x, y) \in \rho(\mathcal{G} \mathbin{\hat{\circ}} \mathcal{F})$, that is,

$$f \circ g(x) = f \circ g(y) \qquad \text{for each } f \in \mathcal{F} \text{ and } g \in \mathcal{G}.$$

Fix $f \in \mathcal{F}$ and let $g \in \overline{\mathcal{G}}$. By definition of the closure, there exists a sequence $(g_n)_{n \in \mathbb{N}} \subseteq \mathcal{G}$ such that $g_n \to g$ uniformly on compact sets. For each $n \in \mathbb{N}$,

$$f(g_n(x)) = f(g_n(y)),$$

and by continuity of $f$ we obtain $f(g(x)) = f(g(y))$. Since $f \in \mathcal{F}$ and $g \in \overline{\mathcal{G}}$ were arbitrary,

$$(x, y) \in \rho(\overline{\mathcal{G}} \mathbin{\hat{\circ}} \mathcal{F}),$$

i.e.,

$$\rho(\mathcal{G} \mathbin{\hat{\circ}} \mathcal{F}) \subseteq \rho(\overline{\mathcal{G}} \mathbin{\hat{\circ}} \mathcal{F}).$$

This proves the first equality.

We now prove the second equality. Since $\mathcal{G} \mathbin{\hat{\circ}} \mathcal{F} \subseteq \mathcal{G} \mathbin{\hat{\circ}} \overline{\mathcal{F}}$, we have

$$\rho(\mathcal{G} \mathbin{\hat{\circ}} \overline{\mathcal{F}}) \subseteq \rho(\mathcal{G} \mathbin{\hat{\circ}} \mathcal{F}).$$

For the reverse inclusion, let $(x, y) \in \rho(\mathcal{G} \mathbin{\hat{\circ}} \mathcal{F})$, that is,

$$f \circ g(x) = f \circ g(y) \qquad \text{for each } f \in \mathcal{F} \text{ and } g \in \mathcal{G}.$$

Fix $g \in \mathcal{G}$ and let $f \in \overline{\mathcal{F}}$. By definition of the closure, there exists a sequence $(f_n)_{n \in \mathbb{N}} \subseteq \mathcal{F}$ such that $f_n \to f$ uniformly on compact sets. For each $n \in \mathbb{N}$,

$$f_n(g(x)) = f_n(g(y)),$$

hence taking limits yields $f(g(x)) = f(g(y))$. Since $g \in \mathcal{G}$ and $f \in \overline{\mathcal{F}}$ were arbitrary,

$$(x, y) \in \rho(\mathcal{G} \mathbin{\hat{\circ}} \overline{\mathcal{F}}),$$

i.e.,

$$\rho(\mathcal{G} \mathbin{\hat{\circ}} \mathcal{F}) \subseteq \rho(\mathcal{G} \mathbin{\hat{\circ}} \overline{\mathcal{F}}).$$

This concludes the proof. $\square$

**Lemma 5.** *Let $M$ be a layer space. Then for each $k, h \in \mathbb{N}_0$ we have*

$$C^{k \times 1} \mathbin{\hat{\circ}} P^{1 \times h} \subseteq P^{k \times h}.$$

*Proof.* Let $\phi \in C^{k \times 1}$. Then $\phi$ can be written as

$$\phi(x_1, \ldots, x_k) = \phi_1(x_1) + \cdots + \phi_k(x_k)$$

for some $\phi_1, \ldots, \phi_k \in C$.
Let $\psi \in P^{1 \times h}$. Then $\psi$ can be written as

$$\psi(x) = \big(\psi_1(x), \ldots, \psi_h(x)\big)$$

for some $\psi_1, \ldots, \psi_h \in P$.
The composition $\psi \circ \phi \in C^{k \times 1} \mathbin{\hat{\circ}} M^{1 \times h}$ can be written

$$\psi \circ \phi(x_1, \ldots, x_k) = \Big(\sum_{j=1}^{k} \psi_i \circ \phi_j(x_j)\Big)_{i,j}$$

where $\psi_i \circ \phi_j \in M$ for each $i$ and $j$ since each $\phi_j$ is a scalar multiple of the identity plus a translational part. $\square$

**Lemma 6.** *The projection $\pi_i^* \, \mathcal{U}_\sigma(C, P, C)$ separates $\mathrm{Stab}_{S_n}(i)$-orbits in $\mathbb{R}^n$.*

*Proof.* Without loss of generality, assume $i = 1$, and note that

$$\pi_1^* \, \mathcal{U}_\sigma(C, P, C) = \mathcal{U}_\sigma(C, P, P_1),$$

where $P_1 := \pi_1^* C$. By Lemma 4, we obtain

$$\bigcup_{k \in \mathbb{N}} \mathcal{N}_\sigma(C^{1 \times k}, C^{k \times 1}) \, \hat{\diamond} \bigcup_{h \in \mathbb{N}} \mathcal{N}_\sigma(P^{1 \times h}, P_1^{h \times 1}) \subseteq \bigcup_{k, h \in \mathbb{N}} \mathcal{N}_\sigma(C^{1 \times k}, P^{k \times h}, P_1^{h \times 1}).$$

Then, by Lemma 5, we have

$$\rho\big(\mathcal{U}_\sigma(C, P, P_1)\big) = \rho\Big( \bigcup_{k, h \in \mathbb{N}} \mathcal{N}_\sigma(C^{1 \times k}, P^{k \times h}, P_1^{h \times 1}) \Big)$$

$$\subseteq \rho\Big( \bigcup_{k \in \mathbb{N}} \mathcal{N}_\sigma(C^{1 \times k}, C^{k \times 1}) \, \hat{\diamond} \bigcup_{h \in \mathbb{N}} \mathcal{N}_\sigma(P^{1 \times h}, P_1^{h \times 1}) \Big)$$

$$= \rho\big(\mathcal{U}_\sigma(C, C) \, \hat{\diamond} \, \mathcal{U}_\sigma(P, P_1)\big).$$

Since functions in $\mathcal{U}_\sigma(C, P, P_1)$ are $\mathrm{Stab}_{S_n}(1)$-invariant, they can at most separate $\mathrm{Stab}_{S_n}(1)$-orbits. Therefore, it suffices to show that $\mathcal{U}_\sigma(C, C) \, \hat{\diamond} \, \mathcal{U}_\sigma(P, P_1)$ separates these orbits. We know by Proposition 5 that

$$\mathcal{U}_\sigma(C, C) = \Big\{ (x_1, \ldots, x_n) \mapsto (f(x_1), \ldots, f(x_n)) \,\Big|\, f \in \mathcal{C}(\mathbb{R}) \Big\}.$$

We now characterize elements in $\mathcal{U}_\sigma(P, P_1)$. Before proceeding, recall that

$$C = \{\, x \mapsto \lambda x + \mu \mathbb{1} \mid \lambda, \mu \in \mathbb{R} \,\}.$$

Then

$$P_1 = \pi_1^* C = \{\, x \mapsto \pi_1(\lambda x + \mu \mathbb{1}) \mid \lambda, \mu \in \mathbb{R} \,\} = \{\, x \mapsto \lambda e_1^\top x + \mu \mid \lambda, \mu \in \mathbb{R} \,\}.$$

Elements in $\mathcal{U}_\sigma(P, P_1)$ are limits of functions of the form

$$\eta(x) = A \circ \tilde{\sigma} \circ B(x),$$

where

$$B = \begin{bmatrix} b_{1,1} \ \mathrm{id} + b_{1,2} \ \mathbb{1}^\top \mathbb{1} \\ \vdots \\ b_{h,1} \ \mathrm{id} + b_{h,2} \ \mathbb{1}^\top \mathbb{1} \end{bmatrix}, \qquad A = \begin{bmatrix} a_1 \cdot e_1^\top & \cdots & a_h \cdot e_h^\top \end{bmatrix},$$

for arbitrary $h \geq 1$. Then, for $x = (x_1, \ldots, x_n)$,

$$\eta(x) = \sum_{r=1}^h a_r \, \sigma\Big(b_{r,1} x_1 + b_{r,2}(x_1 + \cdots + x_n)\Big) = \zeta(x_1, x_1 + \cdots + x_n),$$

where $\zeta \in \mathcal{N}_\sigma(L^{2 \times h}, L^{h \times 1})$. Hence,

$$\mathcal{U}_\sigma(P, P_1) = \Big\{ (x_1, \ldots, x_n) \mapsto f(x_1, x_1 + \cdots + x_n) \,\Big|\, f \in \mathcal{C}(\mathbb{R}^2) \Big\}.$$

Therefore,

$$\mathcal{U} := \mathcal{U}_\sigma(C, C) \, \hat{\diamond} \, \mathcal{U}_\sigma(P, P_1) = \Big\{ (x_1, \ldots, x_n) \mapsto f\big(g(x_1), \, g(x_1) + \cdots + g(x_n)\big)$$

$$\Big|\, f \in \mathcal{C}(\mathbb{R}^2), \, g \in \mathcal{C}(\mathbb{R}) \Big\}.$$

Note that the family

$$\Big\{ (x_1, \ldots, x_n) \mapsto g(x_1) + \cdots + g(x_n) \,\Big|\, g \in \mathcal{C}(\mathbb{R}) \Big\}$$

separates $x = (x_1, \ldots, x_n)$ and $y = (y_1, \ldots, y_n)$ if and only if there exists a permutation $\gamma \in S_n$ such that $x = \gamma y$. Moreover, $\mathcal{U}$ separates $x$ and $y$ whenever $x_1 \neq y_1$, for instance by choosing $g = \mathrm{id}$ and $f(u, v) = u$. Thus, $\mathcal{U}$ separates $x$ and $y$ if and only if there exists a permutation $\gamma \in \mathrm{Stab}_{S_n}(1)$ such that $x = \gamma y$. This concludes the proof.

$\square$

