# OpenReview forum: "On Universality of Deep Equivariant Networks"
_ICLR.cc/2026/Conference — ICLR 2026 Poster_

### Official Review · Reviewer_hN6i · 2025-10-23

**Soundness:** 2
**Presentation:** 2
**Contribution:** 2
**Rating:** 4
**Confidence:** 3

**Summary:**

This paper addresses the theoretical foundations of universality in equivariant neural networks, a topic for which rigorous results remain limited. Existing universality theorems often rely on highly restrictive assumptions—such as regular or higher-order tensor representations that induce prohibitively large hidden dimensions, or architectures that are restricted to the invariant case. The authors develop a more general and unified theoretical framework that encompasses both invariant and equivariant networks. For invariant architectures, they prove a universality theorem under separation constraints, showing that the inclusion of a fully connected readout layer enables universal approximation within the class of separation-constrained continuous functions. For equivariant networks, where existing results are even rarer, the paper identifies a key limitation of standard separability assumptions and introduces the concept of entry-wise separability, a stronger and more suitable criterion. The authors prove that equivariant networks achieve universality within this regime, provided they have sufficient depth or appropriate readout layers. Together with earlier results showing the failure of universality for shallow models, this work highlights depth and readout structure as fundamental mechanisms for universality. The framework also unifies and extends prior specialized results, offering a clearer theoretical understanding of how equivariant architectures can approximate complex functions while maintaining symmetry constraints.

**Strengths:**

This paper tackles a compelling and important direction—building theory for symmetry-aware (equivariant/invariant) neural networks—and does so with clarity and breadth. The authors provide a unified, architecture-agnostic framework that cleanly separates the roles of separation, depth, and readout layers, resolving long-standing confusion about why models with identical separation power can differ in approximation power. The introduction of entry-wise separability is a crisp and genuinely new lens that both diagnoses the failure of standard separability to characterize equivariant universality and supplies the right notion under which strong theorems can be proved. The results are conceptually satisfying and practically meaningful: for invariants, a fully connected readout restores universality within the separation-constrained class; for equivariants, universality emerges either after finite depth stabilization or via specific output layers that act as equivariant analogues of readouts. These statements not only subsume and extend prior specialized results but also give a reassuring message for practice—maximal expressivity is reached at finite depth, avoiding the specter of unbounded “depth chasing.” The presentation is mathematically careful (clear definitions, clean statements, explicit assumptions) and the work advances the field by turning an intuitively appealing idea—that symmetry should guide learning—into a rigorous set of principles that explain when and how symmetry-respecting networks are universal.

**Weaknesses:**

While the paper offers a clean and unified treatment, its contribution risks being perceived as incremental given a substantial body of prior work on universality for symmetry-aware models (invariant and equivariant networks, WL-based expressivity, PointNet/graph settings, and regular-representation proofs). The central message—that depth and readout layers restore universality under separation constraints—resonates with, and in places refines, known intuitions; however, the practical novelty over existing universality theorems may feel limited without sharper separations or new, unexpected consequences. Methodologically, the results hinge on point-wise activations and permutation representations under finite groups, leaving unclear how far the theory extends to continuous/compact Lie groups, non-pointwise nonlinearities, or mixed representation stacks that are common in modern E(3)/SO(3) models. The proposed entry-wise separability is mathematically convenient but may be difficult to verify or estimate in practice, and its relationship to measurable statistics used by practitioners (e.g., tensor orders, message-passing depth, spectral gaps) is not yet operationalized. The universality guarantees are asymptotic and do not provide approximation rates, sample complexity, or robustness to model misspecification—key levers for assessing when added depth/readouts actually help. Moreover, the theory presumes or requires stabilization of separation with depth but offers limited guidance for identifying the stabilization threshold for concrete architectures. Finally, the paper is purely theoretical: without constructive recipes, empirical probes, or optimization analysis, it remains uncertain how the results translate into trainable, parameter-efficient designs under real-world constraints.

**Questions:**

Here are reviewer-style questions that incorporate your points:
1.	Generalization error and sample complexity. Your results are asymptotic in function approximation. Can the framework yield generalization bounds—e.g., via Rademacher or covering-number control of the neural spaces—expressed in terms of the entry-wise separation profile, the depth at which separation stabilizes, and the size/structure of the readout? In particular, can you derive sample complexity or margin-based bounds that certify when added depth/readout improves test error rather than only representation capacity?
2.	Function spaces beyond continuity. Universality is stated over continuous functions. Can you extend the results—or obtain rates—for other spaces such as L^p, Hölder/Sobolev/Besov, BV, or Barron-type spaces? Is entry-wise separability compatible with norms/seminorms in these spaces so that one can state approximation theorems with explicit error decay under smoothness or spectral assumptions?
3.	Connections to integral representations of shallow nets. Classical approximation theory links shallow networks to integral representations (ridgelet transform, Barron spaces, Radon-type decompositions). Is there a formal relationship between your entry-wise separability (or separation stabilization with depth) and the spectral measures that appear in ridgelet/Barron analyses? For instance, can depth plus an appropriate readout be interpreted as enforcing or approximating certain ridgelet spectra or moment constraints, and do your results recover known rate results in Barron regimes or suggest new ones for equivariant settings?
4.	Operational criteria. Practically, how can one estimate or test entry-wise separability (and its stabilization depth) from finite data or from an architecture’s design (e.g., filter classes, tensor orders)? Are there computable proxies (e.g., spectral norms of intertwiners, orbit counts, WL-width analogs) that permit a-priori generalization guarantees or architecture selection?
5.	Robustness and misspecification. Can your theory quantify how approximation/generalization degrades under model misspecification (e.g., when the true target violates the assumed separation relations slightly), or under noisy labels/inputs? Are there stability bounds for the entry-wise notion analogous to Lipschitz-type robustness?
6.	From existence to construction. Your theorems are existential. Do they imply constructive schemes (e.g., explicit readout designs or depth schedules) that achieve provable approximation rates and generalization bounds in the above function spaces, perhaps by leveraging ridgelet-inspired initialization or kernel/NTK limits adapted to entry-wise separability?

---

> ### Author Response · Authors · 2025-11-19
> **Answer to Reviewer hN6i (1/2)**
>
> We appreciate the reviewer’s feedback. In the following, we tackle the concerns mentioned in the review and reply to the individual questions.
>
> > While the paper offers a clean and unified treatment, its contribution risks being perceived as incremental given a substantial body of prior work on universality for symmetry-aware models (invariant and equivariant networks, WL-based expressivity, PointNet/graph settings, and regular-representation proofs). [...]
>
> To the best of our knowledge, the following phenomena are first identified in this work and, in our opinion, are new and somewhat surprising relative to the existing literature:
>
> - First, we show that separation is not sufficient to guarantee universality in the equivariant case (Example 3).
> - Second, we introduce the notion of entry-wise separation and show that, in the equivariant setting with an appropriate readout, this property is sufficient to obtain separation-constrained universality (Theorem 3), in stark contrast with the invariant case (Theorem 1).
> - Finally, our results (Theorems 1 and 2) are not merely generalizations of previously known theorems: the proposed proofs are not incremental adaptations of existing arguments but instead rely on new techniques, such as approximation via neural networks with rational parameters (see Definition 13 and Lemma 2 in Appendix B), which are distinct from those used in the prior literature.
>
> > Methodologically, the results hinge on point-wise activations and permutation representations under finite groups, leaving unclear how far the theory extends to continuous/compact Lie groups, non-pointwise nonlinearities, or mixed representation stacks that are common in modern E(3)/SO(3) models.
>
> We acknowledge the criticism, but we stress that our theorems substantially broaden the scope of several established results in the geometric deep learning literature.
>
> As the reviewer themselves notes in the strengths section, our work “These statements [...] subsume and extend prior specialized results”, which appears to contrast with this concern about limited scope.
>
> Furthermore, our contributions are not limited to extending existing statements: the proofs we introduce are not straightforward adaptations of known arguments, but instead rely on techniques that, to the best of our knowledge, are new in this context.
>
> While the paper intentionally focuses on a specific setting, we believe that its scope is nonetheless substantially broad.
>
> > The proposed entry-wise separability is mathematically convenient but may be difficult to verify or estimate in practice, and its relationship to measurable statistics used by practitioners (e.g., tensor orders, message-passing depth, spectral gaps) is not yet operationalized.
>
> In general, separation, as well as entry-wise separation, is difficult to characterize, but not difficult to compare this property for different models; see Theorem 5 in [1].
> While a full characterization would be preferable, being able to compare the separation power of different models already has substantial empirical impact.
>
>
> > The universality guarantees are asymptotic and do not provide approximation rates, sample complexity, or robustness to model misspecification—key levers for assessing when added depth/readouts actually help. [...]
>
> As is common for work on universality, our contribution is primarily theoretical.
> We agree with the reviewer that experiments would strengthen the paper.
> However, universality cannot be directly controlled by the experimenter; it can only be influenced indirectly by adjusting other factors, such as hyperparameters or activation functions.
> These choices affect not only universality, but also other crucial aspects, such as generalization and trainability, which in turn shape empirical performance.
> As a result, designing experimental settings that isolate the effect of approximation power is challenging, since it is difficult to disentangle the impact of hyperparameter changes on these different components.
>
> It is our intention to further investigate how hyperparameter choices influence these additional aspects.
> Once we have a clearer understanding of their role, we believe it will be possible to design a detailed and meaningful experimental study.
> In line with the last part of the reviewer’s comment, we see our theoretically grounded contribution as an important step toward a more complete understanding of the learning pipeline of equivariant networks, and as a foundation for future theoretical developments that can be validated empirically in a principled way.

---

> > ### Author Response · Authors · 2025-11-19
> > **Answer to Reviewer hN6i (2/2)**
> >
> > > Here are reviewer-style questions that incorporate your points:
> > Generalization error and sample complexity. [...]
> >
> > This is indeed a fundamental topic, but it lies completely outside the scope of the present paper and would require a very different set of tools.
> >
> > We also note that classical complexity measures such as Rademacher complexity and VC dimension are known to be poorly suited for capturing the generalization behavior of overparameterized models such as neural networks, often leading to vacuous generalization bounds (see [2, 3, 4] for general references on this point).
> >
> > Other, more promising generalization bounds are under active development (e.g., [5]), but they are not yet suited to fully address even the simple case of vanilla neural networks trained with SGD.
> > Therefore, we consider a detailed study of the generalization error to be beyond the scope of this work.
> >
> > However, we believe that our results constitute a relevant step toward understanding the role of architectural choices in the generalization of equivariant neural networks trained with SGD, as the study of universality, even in the purely existential setting, provides a necessary condition for learning a given task to arbitrary accuracy.
> >
> >
> > > Function spaces beyond continuity. Universality is stated over continuous functions. Can you extend the results—or obtain rates—for other spaces such as L^p, Hölder/Sobolev/Besov, BV, or Barron-type spaces? Is entry-wise separability compatible with norms/seminorms in these spaces so that one can state approximation theorems with explicit error decay under smoothness or spectral assumptions?
> >
> >
> > > Connections to integral representations of shallow nets. Classical approximation theory links shallow networks to integral representations (ridgelet transform, Barron spaces, Radon-type decompositions). Is there a formal relationship between your entry-wise separability (or separation stabilization with depth) and the spectral measures that appear in ridgelet/Barron analyses? For instance, can depth plus an appropriate readout be interpreted as enforcing or approximating certain ridgelet spectra or moment constraints, and do your results recover known rate results in Barron regimes or suggest new ones for equivariant settings?
> >
> > To the best of our knowledge, Barron spaces and ridgelet-based approaches are defined only for shallow networks.
> > However, as suggested by the title of the paper, our goal is to investigate the role of depth in approximation.
> > Extending these spaces and norms to the deep invariant and equivariant settings would be a remarkable contribution, but we believe them to be out of the scope of the present paper.
> >
> > [1] M. Pacini et al., *Separation Power of Equivariant Neural Networks* \
> > [2] A. G. Wilson, *Deep Learning is Not So Mysterious or Different* \
> > [3] C. Zhang et al., *Understanding deep learning (still) requires rethinking generalization* \
> > [4] M. Belkin, *Reconciling modern machine-learning practice and the classical bias–variance trade-off* \
> > [5] G. K. Dziugaite and D. M. Roy, *Computing nonvacuous generalization bounds for deep (stochastic) neural networks with many more parameters than training data*

---

> > > ### Comment · Reviewer_hN6i · 2025-11-25
> > >
> > > Thank you very much for your thorough and constructive responses to my comments and questions. Your explanations have addressed my concerns satisfactorily, and I now have a much clearer understanding of the contributions and scope of the proposed framework. In light of these improvements and clarifications, I have raised my evaluation of the paper.

---

> > > > ### Author Response · Authors · 2025-11-28
> > > >
> > > > We thank the reviewer for the response, the positive re-evaluation, and the feedback, and we remain open to any further comments.

---

### Official Review · Reviewer_gzFd · 2025-10-29

**Soundness:** 3
**Presentation:** 3
**Contribution:** 3
**Rating:** 8
**Confidence:** 2

**Summary:**

The authors show a separation-constrained universality theorem for invariant networks, and they extend the result to the case of equivariant networks by introducing the notion of entry-wise separability.

**Strengths:**

The authors theoretically investigate separation-constrained universality for invariant and equivariant settings. They introduce a notion called entry-wise separability and show a connection between the depth of the neural networks and the universality. The results are solid and the topic is interesting.

**Weaknesses:**

The notation should be consistent. For example, $h$ represents a natural number in some cases, a function in other cases, and an element in $G$ in other cases. Since this paper focuses on theory, the consistency of the notation improves the readability.

**Questions:**

- In theorem 1, how large can $k$ become? In the standard neural network case, we can obtain arbitrary well-approximated function by increasing the number of the hidden layers. How does $k$ depend on the input dimension and $|G|$?
- Similarly, in Theorem 2, can we determine optimal value of $d$? Since $M$ depends on the functions $\phi^1,\ldots,\phi^k$, are there any relationships between $d$ and $k$?

---

> ### Author Response · Authors · 2025-11-19
> **Answer to Reviewer gzFd**
>
> We thank the reviewer for the feedback. Below, we address the concerns raised and respond to the specific questions.
>
> > The notation should be consistent. For example, $h$ represents a natural number in some cases, a function in other cases, and an element in $G$ in other cases. Since this paper focuses on theory, the consistency of the notation improves the readability.
>
> We agree and thank the reviewer for highlighting this issue. For this reason, we have uploaded a revised version with improved notation and a more streamlined proof of Theorem 1, and we are continuing to refine the manuscript and remain open to further suggestions.
>
> > In theorem 1, how large can $k$ become?
>
> In the proof of Theorem 1, we denote by $k$ the number of hidden units in the last layer.
> This layer must have arbitrary width for the network to approximate the target functions with arbitrary precision, and for certain activations, such as ReLU, this condition is in fact necessary.
> Intuitively, the more complex the target function is, the larger $k$ needs to be.
>
> > In the standard neural network case, we can obtain arbitrary well-approximated function by increasing the number of the hidden layers.
>
> To the best of our knowledge, universality results for families of neural networks with an arbitrary number of layers and fixed width are limited and mostly specific to ReLU activations. In this paper, we do not address this problem and instead align more closely with [1, 2, 3]. Non-trivial depth remains necessary, as shown in Proposition 6 of [4], where it is proved that certain shallow invariant networks fail to achieve separation-constrained universality.
>
> > How does depend on the input dimension and $|G|$?
>
> This is an interesting and relevant question, but we are not able to provide a general answer. However, we can give examples that illustrate the complexity of the landscape and show that the appropriate depth depends strongly on other architectural choices. In particular, the required number of layers depends not only on the cardinality of $G$ or on the input size, but also on the type of hidden representations.
> For instance, if the hidden representation is the regular representation, as in [1], it is possible to achieve universality with a single layer.
> In contrast, for 2-IGNs, due to their equivalence with the 2-WL test, the depth must be at least the number of nodes in the input graph.
> For $k$-IGNs, the number of necessary layers decreases as $k$ increases [2, 3].
>
> > Similarly, in Theorem 2, can we determine optimal value of $d$?
>
> Since separability and entry-wise separability are linked by the reconstruction map (see Appendix C), the examples above also show that determining when $d$ in Theorem 2 stabilizes is a difficult problem, which we expect to be case-specific.
> However, as we explain in the paper, in practice this should not be a major issue, since increasing $d$ has limited computational cost and can be handled during hyperparameter tuning. Our results show that depth beyond a certain limit does not play a role in increasing expressivity, but they do not rule out the possibility that it may influence generalization or the learning dynamics.
>
> > Since depends on the functions $\phi^1,...,\phi^k$, are there any relationships between $d$ and $k$?
>
> Not only is there a relationship between $d$ and $k$, but there is also a dependency on the structure of $M$, or equivalently on the structure of the maps $\phi^i$.
> The examples above show that $d$, $k$, and the structure of $M$ can be complex, likely requiring ad hoc results for each specific architecture.
>
> **References:**
>
> [1] S. Ravanbakhsh. *Universal Equivariant Multilayer Perceptrons* \
> [2] F. Geerts. *The expressive power of kth-order invariant graph networks* \
> [3] H. Maron et al. *Provably Powerful Graph Networks* \
> [4] M. Pacini et al. *On Universality Classes of Equivariant Networks*

---

### Official Review · Reviewer_zQ23 · 2025-10-31

**Soundness:** 2
**Presentation:** 2
**Contribution:** 3
**Rating:** 4
**Confidence:** 4

**Summary:**

This paper develops a general theory for universality in invariant and equivariant neural networks.
For invariant networks, the authors show that adding a fully connected readout layer ensures approximation of all continuous functions consistent with the network’s separation relation.
For equivariant networks, they argue that the usual notion of separation is too weak, introducing the new concept of *entry-wise separation*, which examines separability for each output coordinate.
They then prove two theorems: universality holds (i) once entry-wise separation stabilizes with depth, or (ii) when the network includes a width-1 convolutional readout.
Overall, the paper identifies **depth** and **readout layers** as the key factors controlling universality and provides a unified theoretical framework that connects and extends prior architecture-specific results. :contentReference[oaicite:0]{index=0}

**Strengths:**

- **Originality:** Entry-wise separation is a simple but fresh idea that fixes a real gap in how we reason about equivariant universality. The unifying view across invariant and equivariant settings is valuable.
- **Quality** The Example~3 counterexample is instructive and motivates the new definition well.
- **Clarity:** The big picture is clear: depth stabilizes separation, and readouts enable universal approximation within the correct class. The relation to prior work is well positioned.
- **Significance:** Results inform model design (when to use a readout layer and how deep to go). They also explain why some shallow models fail even if they have the same separation power.

**Weaknesses:**

- **Lack of assumption:** Theorem 1 (and also other theorems) fails, for example, if $\sigma = {\rm id}$.
- **Sparse definitions:** The permutation representation and Eq. (3) are only informally presented. Adding a clear algebraic and matrix-form definition would remove ambiguity.
- **Scope:** The analysis is limited to permutation representations and point-wise activations; potential extensions to other representations are only mentioned briefly.
- **No quantitative results:** The paper proves existence of universality but gives no approximation rates or bounds on the required depth threshold.

**Questions:**

- Could the authors explicitly assume a non-linear activation in the readout MLP and restate Theorem 1 accordingly? Would Theorems 2 and 3 need similar clarification?
- Are there examples showing the exact depth at which entry-wise separation stabilizes for common groups (e.g., $S_n$)?
- Would entry-wise separation remain the correct notion if the representation were not a permutation one (e.g., vector or tensor field representations)?

---

> ### Author Response · Authors · 2025-11-19
> **Answer to Reviewer zQ23 (1/2)**
>
> We are grateful to the reviewer for the feedback. In what follows, we discuss the raised concerns and answer the specific questions.
>
> > Lack of assumption: Theorem 1 (and also other theorems) fails, for example, if $\sigma = id$.
> ​​
>
> > Could the authors explicitly assume a non-linear activation in the readout MLP and restate Theorem 1 accordingly? Would Theorems 2 and 3 need similar clarification?
>
> The reviewer is correct. In particular, we employ the classical result of [1] stating that standard neural networks are universal if and only if their activation functions are non-polynomial. We omitted to state this assumption explicitly in the preliminaries and thank the reviewer for pointing this out.
>
> We uploaded an amended version of the manuscript including this clarification (L205-206).
>
> > Sparse definitions: The permutation representation and Eq. (3) are only informally presented. Adding a clear algebraic and matrix-form definition would remove ambiguity.
>
> We thank the reviewer for the comment. In the revised manuscript, we included further comments on Eq. (3) at the beginning of Appendix B and added Appendix A, where we provide a broader explanation of permutation representations. In particular, we added Proposition 3, which highlights that permutation representations can be realized as matrix representations via permutation matrices.
>
> > Scope: The analysis is limited to permutation representations and point-wise activations; potential extensions to other representations are only mentioned briefly.
>
> We believe that properly treating other architectures requires different tools.
> A popular framework in the geometric deep learning community is that of tensor field networks, which can be viewed as polynomial models; in this setting, the Stone–Weierstrass theorem should be sufficient to obtain universality results.
> Another widely used class of architectures is based on gated nonlinearities and attention mechanisms. To the best of our knowledge, no invariant/equivariant universality results are currently available for such models, and since their structure differs substantially from that of standard neural networks, we expect that proving universality for them would require techniques that are quite different from those employed in our work.
>
> In many practical applications, *continuous* groups are discretized effectively reducing the problem to permutation representations of finite groups, i.e., precisely the setting covered in this paper. Finally, note that, due to [8], pointwise activations can be equivariant only for finite groups and their permutation representations; hence, restricting to permutation representations is a natural choice when working with pointwise activations.
>
> > No quantitative results: The paper proves existence of universality but gives no approximation rates or bounds on the required depth threshold.
>
> We agree that this is a very interesting phenomenon to investigate. \
> That said, obtaining approximation rates under separation constraints appears to be technically challenging. As far as we are aware, the existing works on separation-constrained universality do not provide any quantitative approximation rates [2, 3, 4]. A key difficulty is that one would need to control simultaneously how both depth and width contribute to approximation quality, which is a highly nontrivial problem.
>
> In fact, to the best of our knowledge, approximation-rate results that explicitly depend on depth are currently available only for standard ReLU networks [6], whereas for general activation functions the known results concern shallow architectures [5]. Moreover, it is not obvious that approximation rates are the right tool to understand learning behavior in the overparameterized regime. From our perspective, the main contribution is the existential statement, since it constitutes a necessary condition for learnability.
>
> While a detailed study of approximation rates would certainly enrich the paper, we consider this direction beyond the scope of the present work.

---

> > ### Author Response · Authors · 2025-11-19
> > **Answer to Reviewer zQ23 (2/2)**
> >
> > > Are there examples showing the exact depth at which entry-wise separation stabilizes for common groups (e.g., $S_n$)?
> >
> > In Example 3, we show that entry-wise separation for width-1 convolutions, an $S_n$-equivariant model, stabilizes at depth 2.
> > We could also add that, when the hidden representation is the regular representation (for instance of $S_n$), as in [11], entry-wise separation always stabilizes at depth 2.
> > In contrast, for 2-IGNs, i.e., $S_n$-equivariant models designed to process graphs [10], due to their equivalence with the 2-WL test, the required depth must be at least the number of nodes in the input graph, whereas for $k$-IGNs the necessary number of layers decreases as $k$ increases [2, 3].
> >
> > > Would entry-wise separation remain the correct notion if the representation were not a permutation one (e.g., vector or tensor field representations)?
> >
> > Entry-wise separation is a necessary condition for approximation; hence any model, including polynomial ones, must satisfy suitable separation relations (see [9] for separation results in polynomial models). This is established in the first part of Appendix C, and we will make sure to better highlight this generality in that section.
> >
> > **References:**
> >
> > [1] A. Pinkus. *Approximation theory of the MLP model in neural networks* \
> > [2] H. Maron et al. *Provably Powerful Graph Networks* \
> > [3] F. Geerts. *The expressive power of kth-order invariant graph networks* \
> > [4] F. Geerts and J. L. Reutter. *Expressiveness and Approximation Properties of Graph Neural Networks* \
> > [5] J. Siegel and J. Xu. *Approximation Rates for Neural Networks with General Activation Functions* \
> > [6] J. Siegel. *Optimal Approximation Rates for Deep ReLU Neural Networks on Sobolev and Besov Spaces* \
> > [7] M. Pacini et al. *On Universality Classes of Equivariant Networks* \
> > [8] M. Pacini et al. *A Characterization Theorem for Equivariant Networks with Point-wise Activations* \
> > [9] O. Puny et al. *Equivariant polynomials for graph neural networks* \
> > [10] H. Maron et al. *Invariant and Equivariant Graph Networks*‏ \
> > [11] S. Ravanbakhsh. *Universal Equivariant Multilayer Perceptrons*

---

### Official Review · Reviewer_pabC · 2025-11-01

**Soundness:** 3
**Presentation:** 3
**Contribution:** 2
**Rating:** 4
**Confidence:** 3

**Summary:**

This paper develops a unified theory of separation-constrained universality for deep invariant and equivariant neural networks built from permutation representations and point-wise activations.

* For invariant networks, adding a fully connected readout guarantees universality within the class of continuous functions that respect the model’s separation relation (Theorem 1).
* For equivariant networks, the authors show that standard separation is too coarse and introduce entry-wise separation; universality holds either once entry-wise separation stabilizes with depth (Theorem 2) or by appending a width-1 convolutional readout (Theorem 3). These results recover and generalize several prior architecture-specific theorems and identify depth and readout layers as the decisive mechanisms for universality.

**Strengths:**

* Originality: Introduces entry-wise separation and proves that it exactly characterizes universality classes for broad equivariant architectures---cleanly explaining why standard separation can fail.
* Quality: Precise definitions (layer spaces, neural spaces, universality classes), clear statement of Theorems 1--3, and complete proofs; careful reduction from equivariant to invariant projections via stabilizers.
* Clarity: Running examples (PointNet-style \(P\), convolution \(C\), invariant \(I\)) make abstract statements tangible; Example 3 vividly shows the need for entry-wise separation.
* Significance: Unifies and extends prior results (Maron et al., Ravanbakhsh, Segol & Lipman, Chen et al., Joshi et al.) and translates them into a common separation-constrained viewpoint with practical design levers (depth and readout).

**Weaknesses:**

* Scope limitations: Results are restricted to finite groups and permutation representations with point-wise activations; many modern models rely on continuous groups (e.g., \(E(n)\), \(SO(3)\)) and richer nonlinearities (gates, attention). The authors acknowledge this as future work, but a discussion of which parts of the proofs break (e.g., reconstruction via stabilizers, existence of enough invariant functionals) would sharpen the limits.
* Non-quantitative nature: The theorems are existential; no rates, width bounds, or explicit depth thresholds are provided (Theorem 2 uses a stabilization depth from prior work). Concrete bounds for common groups (e.g., \(S_n\), dihedral groups) and standard blocks would increase practical impact.
* Readout specificity: The readouts that guarantee universality are either fully connected (invariant) or width-1 convolution (equivariant). While natural, it would help to clarify how far this extends to other linear equivariant heads used in practice (e.g., steerable bases, attention-style heads), and whether the “identity-containing” condition on \(M\) can be weakened.
* Examples breadth: Example 3 focuses on width-1 convolutions; complementary examples for graph and manifold settings (beyond the WL context) could better demonstrate the generality of entry-wise separation.

**Questions:**

- Can the authors outline how their proofs might be adapted to yield approximation rates or width/depth trade-offs, even under simplifying assumptions (e.g., Lipschitz target functions on compact domains)?

- Given a working equivariant model, how should one diagnose whether entry-wise separation has stabilized in practice (e.g., via synthetic distinguishability tests), and how does one choose a minimal readout ensuring universality while controlling parameter count?

---

> ### Author Response · Authors · 2025-11-18
> **Answer to Reviewer pabC (1/2)**
>
> We thank the reviewer for the feedback and, in what follows, address the questions and concerns raised in the text.
>
> > Scope limitations: Results are restricted to finite groups and permutation representations with point-wise activations; many modern models rely on continuous groups (e.g., (E(n)), (SO(3))) and richer nonlinearities (gates, attention).
>
> We acknowledge this limitation  explicitly in our paper and note it as  an avenue for future work. Moreover, these models appear to be inherently different from those studied in the present paper, and analyzing their approximation properties would likely require substantially different techniques, constituting a distinct research direction that lies beyond the scope of this work.
>
> However, we would like to emphasize that the presented theorems significantly generalize results from well-regarded work in the geometric deep learning literature. The reviewer, in the strengths section, states that our contribution ``unifies and extends prior results (Maron et al., Ravanbakhsh, Segol & Lipman, Chen et al., Joshi et al.)'', which is at odds with the claim of limited scope.
>
> Moreover, our results are not merely generalizations of previously known theorems: the proposed proofs are not incremental adaptations of existing arguments but instead rely on new techniques, for example approximation via neural networks with rational parameters (see Definition 13 and Lemma 2 in Appendix B), which are distinct from those used in the prior literature.
>
> While we acknowledge that the scope of the paper is focused, we believe it is far from being narrow.
>
> > The authors acknowledge this as future work, but a discussion of which parts of the proofs break (e.g., reconstruction via stabilizers, existence of enough invariant functionals) would sharpen the limits.
>
> Reconstruction via stabilizers and the existence of invariant functions directly extend all results on the density of equivariant models, as shown in the first part of Appendix C.
> We will make sure to better highlight this generality in that section.
>
> The proofs of Theorems 1, 2, and 3 are tailored to neural networks with pointwise activations. In particular, we employ the classical universality result by Pinkus [1].
>
> At first, we believe that, to properly tackle other nonlinearities (which enable the construction of $E(n)$- and $SO(3)$-equivariant models), different tools should be employed.
> For tensor field networks, one could rely on polynomial approximation, and hence a Stone–Weierstrass-type theorem should suffice.
> For gated nonlinearities and attention mechanisms, to the best of our knowledge, there are no invariant/equivariant universality results, and since these models are quite different from standard neural networks, we expect that their universality proofs will require techniques that are substantially different from those used for standard neural networks.
> In other cases, groups such as $SO(3)$ are discretized, often to icosahedral symmetries, thereby reducing the setting to permutation representations of finite groups, which are precisely the cases covered in the present paper.
>
> Finally, note that, due to [8], pointwise activations can be equivariant only for finite groups and their permutation representations; hence, restricting to permutation representations is a natural choice when working with pointwise activations.
>
> > Can the authors outline how their proofs might be adapted to yield approximation rates or width/depth trade-offs, even under simplifying assumptions (e.g., Lipschitz target functions on compact domains)?
>
> > Non-quantitative nature: The theorems are existential; no rates, width bounds, or explicit depth thresholds are provided (Theorem 2 uses a stabilization depth from prior work). Concrete bounds for common groups (e.g., (S_n), dihedral groups) and standard blocks would increase practical impact.
>
> We agree that this would be a particularly interesting phenomenon to study. However, deriving approximation rates in a separation-constrained setting is particularly difficult. To the best of our knowledge, existing results on separation-constrained universality do not provide approximation rates [2, 3, 4]. We believe this difficulty stems from the need to simultaneously understand the role of both depth and width in approximation, which is a rather challenging problem.
>
> Indeed, to the best of our knowledge, approximation-rate results that explicitly involve depth are available only for standard ReLU neural networks [6], whereas results for arbitrary activations are restricted to shallow networks [5].
>
> Moreover, it is not clear that approximation rates are a useful notion for understanding learning in the overparameterized regime. In our view, the existential result is the most important part, since it provides a necessary condition for learnability.
>
> While establishing approximation rate bounds would certainly strengthen the contribution, we believe this to be out of scope for the present work.

---

> ### Author Response · Authors · 2025-11-19
> **Answer to Reviewer pabC (2/2)**
>
> > Readout specificity: The readouts that guarantee universality are either fully connected (invariant) or width-1 convolution (equivariant). While natural, it would help to clarify how far this extends to other linear equivariant heads used in practice (e.g., steerable bases, attention-style heads), and whether the “identity-containing” condition on (M) can be weakened.
>
> Steerable bases are not compatible with pointwise activations, hence they are typically paired with tensor products or gated nonlinearities. For tensor-product nonlinearities, one could rely on polynomial approximation, so Stone-Weierstrass-type theorem should suffice. For gated nonlinearities, to the best of our knowledge, there are no universality results for such architectures. We believe that properly addressing these models would require ad hoc tools, which makes this direction out of scope for the present paper.
>
> The issue raised by the “identity-containing’’ condition is indeed interesting. The separation stabilization theorem assumes this condition, and since our results hinge on this theorem, we also have to impose it. However, in our view, separation could in principle stabilize even without this assumption, although we are not aware of models in the literature that do not satisfy the identity condition.
>
> > Examples breadth: Example 3 focuses on width-1 convolutions; complementary examples for graph and manifold settings (beyond the WL context) could better demonstrate the generality of entry-wise separation.
>
> We use width-1 convolutions in Example 3 to present the simplest working construction that illustrates why the notion of entry-wise separation is necessary and to introduce the concept with minimal notation.
>
> We agree with the reviewer that adding a more practically relevant example would make the exposition more engaging.
>
> In this direction, we could enrich Example 3 by recalling Lemma 5 (see appendix) to more clearly highlight the role of entry-wise separation for permutation equivariant networks.
>
> If we correctly interpret the reviewer's use of ``manifold setting'' as referring to architectures in which neural networks are defined on Riemannian manifolds, such as in [9], we also regard this as future work. For these kinds of networks, to the best of our knowledge, it is not yet clear how to rigorously construct and define equivariant neural networks even at a purely theoretical level, and understanding their universality properties, while certainly interesting, is far beyond the scope of the present paper.
>
> > Given a working equivariant model, how should one diagnose whether entry-wise separation has stabilized in practice (e.g., via synthetic distinguishability tests), and how does one choose a minimal readout ensuring universality while controlling parameter count?
>
> While in practice it is certainly important to understand the depth and width necessary to obtain adequate expressivity for a given task (subject to known theoretical constraints). Such a question can be theoretically addressed by approximation rates or empirically explored by extensive experimentation over different benchmarks.  However, due to the scale and differences, these questions are usually addressed separately from the approximation theory works establishing density and upper bounds on expressiveness.
>
> In practice, it is not necessary to explicitly check stabilization or to control the parameter count. Indeed, the computational cost of the model grows linearly with depth, which makes it feasible to empirically test whether its expressivity is sufficient for the task at each depth, for example via overfitting experiments. Our results show that, once this expressive barrier is reached, improving expressivity requires resorting to more complex hidden representations, which, however, may lead to an intractable computational burden.
> Could the reviewer clarify what is meant by a “synthetic distinguishability test’’ in this context?
>
> **References:**
>
> [1] A. Pinkus. *Approximation theory of the MLP model in neural networks* \
> [2] H. Maron et al. *Provably Powerful Graph Networks* \
> [3] F. Geerts. *The expressive power of kth-order invariant graph networks* \
> [4] F. Geerts and J. L. Reutter. *Expressiveness and Approximation Properties of Graph Neural Networks* \
> [5] J. Siegel and J. Xu. *Approximation Rates for Neural Networks with General Activation Functions* \
> [6] J. Siegel. *Optimal Approximation Rates for Deep ReLU Neural Networks on Sobolev and Besov Spaces* \
> [7] M. Pacini et al. *On Universality Classes of Equivariant Networks* \
> [8] M. Pacini et al. *A Characterization Theorem for Equivariant Networks with Point-wise Activations* \
> [9] A. Kratsios and E. Bilokopytov, *Non-Euclidean Universal Approximation*

---

### Author Response · Authors · 2025-11-19

We thank the reviewers for their reports and address their comments below. In the meantime, we have uploaded a first revised version of the manuscript (changes in teal). We have already implemented many of the reviewers’ suggestions and will continue to refine the manuscript during the discussion period, remaining open to further criticism and constructive feedback.

---

### Meta-Review · Area_Chair_aF1J · 2026-01-07

**Summary:**

This paper develops and proposes a unified theoretical framework of separation‑constrained universality for invariant and equivariant neural networks with permutation representations and point‑wise, non‑polynomial activations. Its key contributions are: (i) a general universality theorem for invariant networks once a fully connected readout is appended (Theorem 1); (ii) the introduction of entry‑wise separation as the right notion for characterizing universality in the equivariant setting; and (iii) two results showing that equivariant networks achieve universality either after depth‑driven stabilization of separation (Theorem 2/Corollary 1) or by adding a width‑1 convolutional readout (Theorem 3).

**Reviewer Concerns:**

Concerns included:

a)  Scope (finite groups/point‑wise activations), lack of quantitative bounds (rates/width/depth), readout generality beyond fully connected / width‑1 conv, and breadth of examples

b) incremental contributions

c) missing explicit assumption of non‑polynomial activations, sparse definitions of permutation representations, scope limited to permutation reps/point‑wise activations, and no quantitative rates/depth thresholds

The above is a non-exhaustive list of concerns, but the authors made several changes and addressed the reviewers' concerns.

They clarified (and amended the manuscript) that non‑polynomial activations are assumed, citing classical universality results (Pinkus) and updating the preliminaries.

They added matrix‑form descriptions and a Proposition 3 showing realisation via permutation matrices, and further comments on Eq. (3) to improve algebraic transparency.

They explained why extending beyond point‑wise activations and finite‑group permutation reps likely requires different tools (e.g., Stone–Weierstrass for polynomial/tensor‑product nonlinearities; gated/attention architectures lacking known universality results), while noting that many practical pipelines discretise continuous groups to finite settings.

**Reviewer Scores:**

I believe reviewers would have responded positively to the rebuttal, and in cases where a reviewer has not already indicated they have revised their evaluation, I believe it would have resulted in revised scores.

---

### Decision · Program_Chairs · 2026-01-26

Accept (Poster)